# UniAPL: A Unified Adversarial Preference Learning Framework for Instruct-Following

## Abstract

Shaping the behavior of powerful Large Language Models (LLMs) to be both beneficial and safe is the central challenge of modern AI alignment. We posit that the post-training alignment process is fundamentally a unified challenge of Preference Learning, encompassing two distinct modalities: learning from demonstrated preferences (e.g., Supervised Fine-Tuning, SFT) and from comparative preferences (e.g., Reinforcement Learning, RL). The current industry-standard pipeline, which processes these preference types sequentially, is inherently flawed due to a critical distributional mismatch between the static expert data and the dynamic policy. This creates two interconnected problems: (1) Offline SFT trains on a fixed expert distribution, but as the policy's own generation distribution drifts, the learned knowledge becomes brittle and unreliable. (2) Subsequent online RL explores to improve generalization, but it operates without direct access to the rich, ground-truth knowledge within the expert demonstrations, making its exploration inefficient and ungrounded. This fundamental separation prevents the two data sources from synergistically regularizing each other. To resolve this, we first reframe alignment as a constrained optimization problem. We then propose Unified Adversarial Preference Learning (UniAPL), a novel framework that directly operationalizes this theory by dynamically bridging the gap between the policy's distribution and the expert's distribution. The ultimate expression of our framework is a simplified, single-stage unified training objective. This approach cohesively learns from mixed batches of SFT and preference feedback data, allowing the dense expert data to directly ground and regularize the online exploration process in every gradient update. This concurrent optimization inherently mitigates the distributional mismatch and maximizes data synergy. We empirically validate our approach on instruction-following tasks using `Qwen3-235B-Instruct-2507` as the expert teacher. Our model demonstrates comparable or superior general capabilities in English, coding, mathematics, and Chinese, while significantly enhancing instruction-following ability; it surpasses the strong GRPO baseline by 5.77% on `Qwen3-0.6B`—matching a 4B model's performance—and exceeds 3.75% on `Qwen3-4B`, even outperforming the teacher model. Furthermore, analysis of response length and log-probability (logp) distributions shows that models trained with UniAPL not only achieve stronger performance but also generate outputs closely resembling expert demonstrations.

## 1 Introduction

Large Language Models (LLMs) represent a paradigm shift in artificial intelligence, demonstrating remarkable capabilities in complex reasoning, content generation, and human-computer interaction Yang et al. (2024); Bubeck et al. (2023). This transformative potential, however, presents a double-edged sword. As these models grow in power and autonomy, ensuring their behavior remains aligned with human values and intent is not merely a technical refinement but the critical frontier of AI system Sun et al. (2025). The pursuit of Artificial General Intelligence (AGI) hinges on our ability to create systems that are not only intelligent but also helpful, harmless, and honest. Post-training alignment has thus emerged as a foundational field of research, dedicated to shaping these powerful models into reliable partners in human progress Tie et al. (2025).

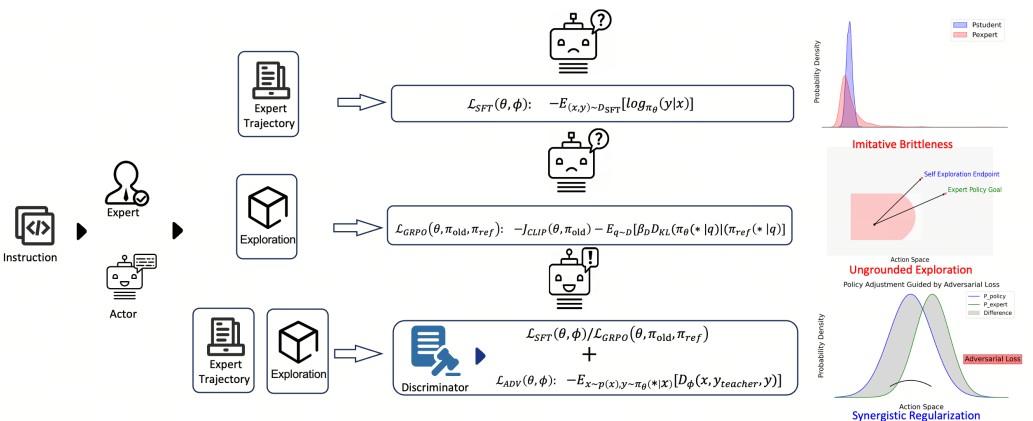

Figure 1: Overview of the UniAPL training framework. UniAPL unifies offline supervised data and online preference signals into a single, coherent optimization process.

| Feature | Non-Adversarial | | | Adversarial(Ours) | | |
|---|---|---|---|---|---|---|
| | SFT | GRPO | SFT→GRPO | A-SFT | A-GRPO | UniAPL |
| Offline Question | ✓ | ✓ | ✓ | ✓ | ✓ | ✓ |
| Offline Response | ✓ | ✗ | ✓ | ✓ | ✓ | ✓ |
| Online Explore | ✗ | ✓ | ✓ | ✓ | ✓ | ✓ |
| Reward Constraints | ✗ | ✓ | ✓ | ✗ | ✓ | ✓ |
| Token Level | ✓ | ✓ | ✓ | ✓ | ✓ | ✓ |
| Sentence Level | ✗ | ✗ | ✗ | ✓ | ✓ | ✓ |

Table 1: Illustration of the differences across training paradigms in data usage, exploration strategies, reward formulation, and loss granularity.

In this work, we introduce a more principled perspective on this challenge: post-training alignment is a unified process of **Preference Learning**. Human intent is conveyed to models through two primary channels of preference data. The first is **Demonstrated Preferences**, where experts provide high-quality examples of desired outputs. Supervised Fine-Tuning (SFT) is the mechanism for learning from this data, instilling the model with foundational knowledge, style, and safety protocols—what we term *Imitative Information* Ouyang et al. (2022). The second is **Comparative Preferences**, where humans rate or rank different model outputs Christiano et al. (2017). Reinforcement Learning (RL) techniques like Direct Preference Optimization (DPO) Rafailov et al. (2023) and Group Relative Policy Optimization (GRPO) Shao et al. (2024) are used to learn from this feedback, refining the model's helpfulness and harmlessness.

The fundamental flaw in current methodologies is the **sequential application** of these learning processes, which creates a critical **distributional mismatch** between the static expert data and the dynamic, evolving policy. This mismatch manifests as a severe dilemma with two interconnected problems:

- **The Offline Learning Problem (Imitative Brittleness):** SFT performs offline learning on a fixed, static expert data distribution ($P_{\text{expert}}$). While this provides a crucial knowledge base, it risks overfitting Liu et al. (2021). As the policy begins to generate its own responses, its underlying distribution ($P_{\text{policy}}$) inevitably diverges from the expert's Yu et al. (2023). Due to this gap, the knowledge learned via SFT becomes brittle and fails to generalize reliably to out-of-distribution scenarios.

- **The Online Learning Problem (Ungrounded Exploration):** Subsequent online RL aims to improve generalization by exploring beyond the static expert data. However, this exploration is ungrounded. The policy operates based on its own distribution ($P_{\text{policy}}$) but lacks real-time, direct access to the dense, ground-truth knowledge contained within the expert demonstrations. This makes its exploration inefficient, prone to catastrophic forgetting, and susceptible to unconstrained policy drift towards reward-hacking solutions Wang et al. (2024a); Carta et al. (2023).

This fundamental separation of offline grounding and online exploration prevents the two data sources from synergistically regularizing each other, trapping the alignment process in a state of inefficiency and instability. To resolve this dilemma, we introduce **Unified Adversarial Preference Learning (UniAPL)**, a new paradigm that directly operationalizes a more robust theoretical foundation where alignment is treated as a single, constrained optimization problem. We summarize the key differences in how our method utilizes data, rewards, exploration and loss granularity in Table 1. Our primary contributions are summarized as follows:

- **We re-conceptualize the core challenge of alignment** from simple information decay to a more fundamental problem of distributional mismatch between the brittle, offline-trained policy and the ungrounded, online-exploring policy.

- **We propose a novel framework, UniAPL**, which utilizes a distributional adversarial objective as a dynamic bridge to close this gap, enabling synergistic regularization between demonstrated and comparative preference learning, as illustrated in Figure 1.

- **We introduce a practical, single-stage training paradigm** that cohesively optimizes a unified objective on mixed-data batches. This approach inherently prevents ungrounded drift, maximizes data synergy, and simplifies the entire alignment workflow.

- **We provide extensive empirical validation** for our framework, demonstrating that our unified approach achieves state-of-the-art performance. In particular, we observe absolute improvements of up to 5.77% on `Qwen3-0.6B` and 3.75% on `Qwen3-4B` over strong GRPO baselines.

## 2 RELATED WORKS

**Supervised Fine-Tuning** has become the cornerstone of post-training for LLMs due to its simplicity and effectiveness. However, by minimizing cross-entropy loss through maximum likelihood estimation, SFT tends to overfit training data, resulting in overconfidence and degraded generalization. To address these limitations, researchers have explored regularization strategies Liu et al. (2024c); Li et al.; Pereyra et al. (2017) and high-quality data synthesis methods Taori et al. (2023); Cui et al. (2023); Chen et al. (2016); Maosongcao et al. (2025); Lin et al. (2024) to reduce excessive memorization and knowledge forgetting. More recently, reinforcement learning (RL) has been integrated into the post-training pipeline for its exploratory advantages, and this combination has evolved into the dominant "SFT-then-RL" paradigm, which is now widely adopted in training state-of-the-art LLMs Yang et al. (2025); Liu et al. (2024a); Team et al. (2024); Achiam et al. (2023); Cheng et al. (2023) to balance core capability grounding with enhanced adaptability and robustness.

**Reinforcement Learning** plays a critical role in Post-Training by aligning model outputs with human preferences. Recent work has applied reinforcement learning to domains with verifiable solutions, such as mathematics and programming, leading to reinforcement learning with verifiable rewards (RLVR) and notable performance gains Lambert et al. (2024); Guo et al. (2025); Mroueh (2025). However, the effectiveness of RL is limited by the model's reliance solely on its own output distribution, without access to external knowledge. On challenging instances, models struggle to produce correct answers, and repeated sampling yields little benefit, leading to suboptimal performance. While approaches such as curriculum learning Liu et al. (2024b) and dynamic sampling Yu et al. (2025) have been proposed to mitigate these issues, their effectiveness remains limited. Recently, incorporating external expert knowledge provides a more straightforward and effective ways to address these difficult cases.

**Incorporating external knowledge into reinforcement learning** has been shown to improve training efficiency Chu et al. (2025); Hong et al. (2024); Zhang et al. (2025a). In the exploration phase of RL, repeated failures on difficult problems indicate that they exceed the model's current ability. In such cases, external guidance can mitigate inefficient exploration on hard problems. Previous studies have leveraged high-quality external responses, either by incorporating them into the rollout process Yan et al. or by providing them as exemplars to guide the policy model Yan et al. (2025); Zhang et al. (2025b). However, the former breaks the on-policy nature of RL, while the latter introduces state inconsistencies between exploration and learning process, leading to suboptimal performance. Some studies Fu et al. (2025); Zhang et al. (2025a) introduce cross-entropy loss into RL, but token-level likelihood maximization reduces entropy and ultimately hampers RL effectiveness. Unlike prior approaches, we employ an adversarial framework where the discriminator enforces indistinguishability between policy outputs and expert demonstrations, and we employ the

adversarial framework where the discriminator provides a dynamic, gradient-based regularization signal that guides the student policy towards the semantic manifold of the expert distribution.

## 3 METHODOLOGY: A UNIFIED FRAMEWORK FOR ALIGNMENT

### 3.1 THEORETICAL FOUNDATION: THE ALIGNMENT DUALITY AND ITS DILEMMA

The objective of LLM alignment is to find an optimal policy $\pi_\theta$ that reflects human intent, conveyed through *Imitative Information* from expert demonstrations $D_{\text{SFT}}$ and *Preference Information* from feedback $D_{\text{PREF}}$. An optimally aligned policy must jointly satisfy the constraints from both sources.

**Lemma 1** (Optimality Condition for Alignment). *An optimal aligned policy $\pi_{aligned}$ is the solution to the following constrained optimization problem:*

$$\pi_{aligned} = \arg\max_{\pi_\theta} \mathbb{E}_{x \sim p(x), y \sim \pi_\theta(\cdot|x)}[R_\psi(y|x)] \quad subject\ to \quad D(\pi_\theta||\pi^*) \leq \epsilon \tag{1}$$

*where $R_\psi$ is the reward function from Preference Information, and the constraint, where $D$ is a divergence measure such as KL-divergence, forces the policy $\pi_\theta$ to remain faithful to the expert policy $\pi^*$. The sequential "SFT-then-RL" pipeline can be viewed as a specific, often sub-optimal, strategy for solving this problem by decoupling the objectives and optimizing them sequentially.*

### 3.2 THE UNIAPL FRAMEWORK: AN ADVERSARIAL BRIDGE FOR THE DISTRIBUTIONAL GAP

UniAPL operationalizes the objective in Lemma 1 by introducing a rich, distributional learning signal. This signal is not merely a component of a loss function, but a distinct gradient vector that guides the optimization process.

**The Adversarial Gradient Signal.** We employ a discriminator, $D_\phi$, to enforce distributional consistency. Given a prompt $x$, a student response $y_s \sim \pi_\theta(\cdot|x)$, and a teacher response $y_t \sim \pi_{\text{teacher}}(\cdot|x)$, the discriminator outputs a similarity score $D_\phi(x, y_t, y_s) \in [-1, 1]$. The student policy is trained to maximize this score. This process is driven by the adversarial loss $\mathcal{L}_{\text{ADV}}$:

$$\mathcal{L}_{\text{ADV}}(\theta, \phi) = -\mathbb{E}_{x \sim p(x), y \sim \pi_\theta(\cdot|x)}[D_\phi(x, y_{\text{teacher}}, y)]$$

This loss function generates a corresponding **adversarial gradient**, $\mathbf{g}_{\text{ADV}} = \nabla_\theta \mathcal{L}_{\text{ADV}}(\theta, \phi)$, which acts as a vector in the parameter space, pulling the student policy towards the teacher's semantic manifold.

### 3.3 GRADIENT FORMULATIONS OF ALIGNMENT COMPONENTS

Our unified framework views each alignment stage through the lens of the gradient it contributes to the overall policy update. Appendix A.4 provides a discussion of DPO and KTO.

**Adversarial Supervised Fine-Tuning (A-SFT).**

OBJECTIVE. To produce an update direction that simultaneously mimics expert data and matches its underlying distribution.

METHODOLOGY. Standard SFT is driven by the negative log-likelihood loss, $\mathcal{L}_{\text{SFT}}(\theta) = -\mathbb{E}_{(x,y) \sim D_{\text{SFT}}}[\log \pi_\theta(y|x)]$, which yields an **imitation gradient**, $\mathbf{g}_{\text{SFT}} = \nabla_\theta \mathcal{L}_{\text{SFT}}(\theta)$. A-SFT enhances this by incorporating the adversarial signal. The total gradient for an A-SFT step is a composite vector:

$$\mathbf{g}_{\text{A-SFT}}(\theta, \phi) = \nabla_\theta \mathcal{L}_{\text{A-SFT}}(\theta, \phi) = \mathbf{g}_{\text{SFT}}(\theta) + \lambda_{\text{adv}}\mathbf{g}_{\text{ADV}}(\theta, \phi)$$

**Adversarial Group Relative Policy Optimization (A-GRPO).**

OBJECTIVE. To produce an update direction that seeks higher-reward regions while being grounded by the expert's distribution.

METHODOLOGY. This stage utilizes an online RL algorithm, GRPO, which is driven by a reward model $RM_\psi$ trained on $D_{\text{PREF}}$. For a given prompt, $G$ outputs are sampled from a behavior policy $\pi_{\text{old}}$, and their advantages $\hat{A}_i$ are calculated based on normalized rewards. The GRPO update is derived from its PPO-style loss function, $\mathcal{L}_{\text{GRPO}}$, which combines a clipped surrogate objective $\mathcal{J}_{\text{CLIP}}$ and a KL penalty. This loss is defined as:

$$\mathcal{L}_{\text{GRPO}}(\theta, \pi_{\text{old}}, \pi_{\text{ref}}) = -\left(\mathcal{J}_{\text{CLIP}}(\theta, \pi_{\text{old}}) - \mathbb{E}_{q \sim D}[\beta_D D_{\text{KL}}(\pi_\theta(\cdot|q)||\pi_{\text{ref}}(\cdot|q))]\right)$$

This loss yields a **preference-seeking gradient**, $\mathbf{g}_{\text{GRPO}} = \nabla_\theta \mathcal{L}_{\text{GRPO}}$. A-GRPO augments this with the adversarial signal. The total gradient for an A-GRPO step is therefore:

$$\mathbf{g}_{\text{A-GRPO}}(\theta, \phi) = \nabla_\theta \mathcal{L}_{\text{A-GRPO}} = \mathbf{g}_{\text{GRPO}}(\theta) + \lambda_{\text{adv}} \mathbf{g}_{\text{ADV}}(\theta, \phi)$$

### 3.4 THE UNIFIED TRAINING OBJECTIVE: A SINGLE-STAGE PARADIGM

The ultimate expression of UniAPL is a single-stage, unified training objective whose power is best understood by analyzing its gradient. We provide further discussion in appendix A.1 and A.3.

**Methodology.** We construct mixed batches from $D_{\text{SFT}}$ and $D_{\text{PREF}}$ and optimize a single loss function:

$$\mathcal{L}_{\text{Unified}}(\theta, \dots) = \alpha \cdot \mathcal{L}_{\text{A-SFT}}(\theta, \phi) + (1 - \alpha) \cdot \mathcal{L}_{\text{A-GRPO}}(\theta, \dots) \tag{2}$$

**The Unified Gradient.** The essence of our framework is revealed in the gradient of this unified loss, which dictates every parameter update. The total gradient, $\mathbf{g}_{\text{Unified}}$, is a weighted sum of the gradients from the A-SFT and A-GRPO components, computed on their respective data slices within the mixed batch:

$$\mathbf{g}_{\text{Unified}} = \nabla_\theta \mathcal{L}_{\text{Unified}} = \alpha \cdot \mathbf{g}_{\text{A-SFT}}|_{\text{SFT}} + (1 - \alpha) \cdot \mathbf{g}_{\text{A-GRPO}}|_{\text{PREF}} \tag{3}$$

By substituting the definitions of the component gradients, we can see that the final update vector is a synergistic combination of four fundamental learning signals:

$$\mathbf{g}_{\text{Unified}} = \underbrace{\alpha \nabla_\theta \mathcal{L}_{\text{SFT}}}_{\substack{\text{Imitation Signal} \\ \text{(from SFT data)}}} + \underbrace{(1 - \alpha) \nabla_\theta \mathcal{L}_{\text{GRPO}}}_{\substack{\text{Preference Signal} \\ \text{(from RL data)}}} + \underbrace{\lambda_{\text{adv}} \left(\alpha \nabla_\theta \mathcal{L}_{\text{ADV}}|_{\text{SFT}} + (1 - \alpha) \nabla_\theta \mathcal{L}_{\text{ADV}}|_{\text{PREF}}\right)}_{\substack{\text{Global Distributional} \\ \text{Regularization}}}$$

This equation mathematically demonstrates that every single parameter update is simultaneously pushed by gradients for imitation, preference-seeking, and distributional grounding. This is the core of our unified and synergistic optimization process.

**Advantages of the Unified Paradigm.** By architecting the training process around this single, synergistic gradient, our methodology directly mitigates the core challenges of the sequential paradigm, yielding three distinct advantages:

1. **Inherent Prevention of Ungrounded Policy Drift.** The concurrent training on both SFT and preference data ensures the expert distribution is not a forgotten starting point but a live, dynamic regularizer. The policy is constantly anchored to this ground-truth data manifold, effectively preventing the ungrounded and unchecked policy drift that plagues a separate RL stage.

2. **Synergistic Data Utilization for Enhanced Generalization.** This is more than just data efficiency; it is data synergy. The model learns to balance imitation and preference optimization in every gradient update. The RL signal pushes the model to generalize beyond the potentially overfitted SFT data, while the SFT data provides a rich, grounding signal that makes RL updates more stable and sample-efficient.

3. **Simplified and Efficient Training Workflow.** Beyond the theoretical benefits, our unified approach offers significant operational advantages. The complex, multi-stage training, validation, and model hand-off process is replaced by a single, continuous training run. This dramatically simplifies the overall alignment workflow, reducing engineering overhead and potential sources of error.

These benefits, which we will demonstrate empirically in the following sections, establish our unified approach as a more robust, efficient, and conceptually sound paradigm for LLM alignment.

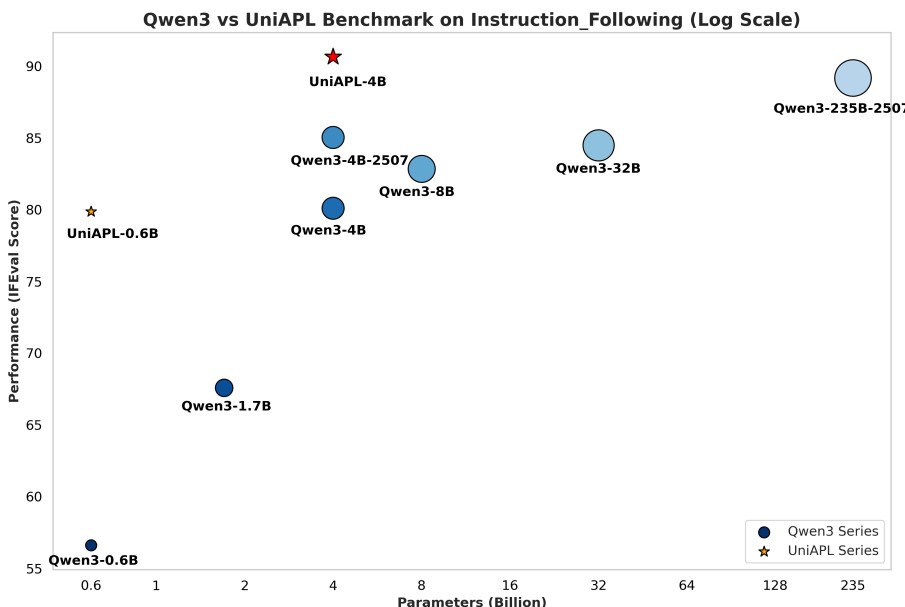

Figure 2: UniAPL performance on the Instruction-Following benchmark (average of IFEVAL and MultiIF). Notably, `UniAPL-0.6B` performs on par with the much larger `Qwen3-4B` model, and `UniAPL-4B` outperforms its teacher model, `Qwen3-235B-A22B-Instruct-2507`, demonstrating the effectiveness and efficiency of the UniAPL approach even with significantly smaller model sizes.

## 4 EXPERIMENTS

In this section, we evaluate the performance of UniAPL on instruction-following tasks and conduct a comprehensive analysis of the model's output behavior.

### 4.1 EXPERIMENTAL SETUP

**Datasets.** We use three instruction-following datasets—AutoIF Dong et al. (2024), IFevallike Xu et al. (2024), and IFBench Pyatkin et al. (2025)—as our training data, with sizes of 61k, 56k, and 38k examples, respectively. The 61k and 56k examples come from the original AutoIF and IFevallike datasets, while the 38k GT responses in IFBench are obtained by filtering the original 95k examples using a forward pass of a `Qwen2.5-72B` model to retain only those that successfully pass the verification function. Each example consists of a prompt, a verification function, and a corresponding ground-truth response (GT) that passes the verification function. Examples of the training data, validation functions, and response data can be found in the appendix B.5.

**Teacher Policy Responses.** The teacher policy responses are generated by performing forward inference with the `Qwen3-235B-Instruct-2507` model on the prompts from the datasets. We also include an ablation study that uses `Qwen3-32B` as the teacher model, which is provided in B.2. Note that these responses are not guaranteed to pass the verification function.

**Benchmarks.** To verify the effectiveness of our method,we evaluate it on two common instruction-following benchmarks and ten general-purpose tasks,including (IFEval Zhou et al. (2023), MultiIF He et al. (2024)), general English (MMLU Hendrycks et al. (2020), MMLU-Pro Wang et al. (2024b), GPQA-Diamond Rein et al. (2024)), coding(Humaneval Chen et al. (2021), Mbpp Austin et al. (2021)) and mathematics (GSM8K Cobbe et al. (2021), Math-500 Lightman et al. (2023), TheoremQA Chen et al. (2023)), and Chinese (CMMLU Li et al. (2023), CEval Huang et al. (2023)). We report the average performance per domain in the main tables, with detailed results for each dataset provided in the Appendix B.4.

**Setting.** According to our theoretical insight, a greater discrepancy between the teacher and student policies is beneficial for performance; To validate the generality and effectiveness of UniAPL, we incorporate it into both supervised fine-tuning (SFT) and reinforcement learning (RL) settings,

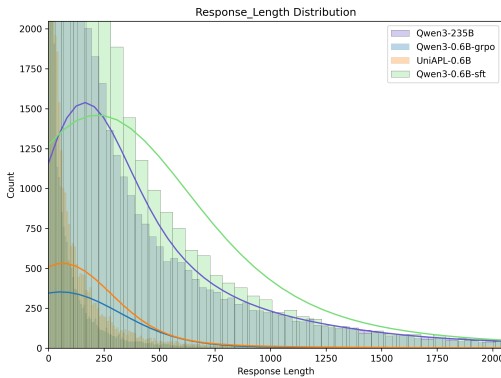 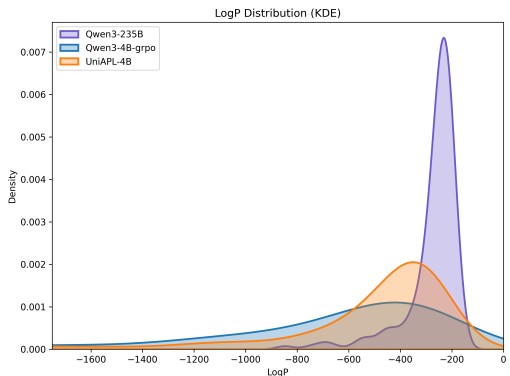

Figure 3: Response length count distributions on the IFBench-38K dataset under teacher models and different training paradigms. UniAPL achieves a response length distribution more consistent with the teacher than GRPO.

Figure 4: Kernel Density Estimation (KDE) of log-probability differences between student and teacher models trained with GRPO and UniAPL. UniAPL exhibits a narrower distribution and reduced divergence, indicating closer alignment with the teacher.

training Qwen3 models of two sizes: 0.6B and 4B. In the SFT setting, we directly compute the loss using standard cross-entropy between predictions and ground-truth responses. For RL, we directly adopt GRPO. For the verifiable reward, the model receives a reward of 1 if its response passes the verification function,and 0 otherwise. We evaluate the impact of integrating UniAPL at different stages: A-SFT denotes the addition of the UniAPL adversarial loss to the cross-entropy loss, while A-GRPO represents the incorporation of the adversarial loss into the GRPO objective. When computing distances using UniAPL, to avoid introducing a discriminator that would be progressively updated during training, we adopt and modify POLAR Dou et al. (2025) as a universal discriminator. POLAR is a reference-based reward model that evaluates the likelihood that a response and its reference originate from the same policy model and outputs a corresponding score. The details of our modifications to adapt POLAR into a universal discriminator are provided in Appendix A.3. Additionally, we conduct an ablation study using BGE-M3 as the discriminator, and the results are reported in B.4. Finally, we use CHORD Zhang et al. (2025a), a method that jointly performs supervised fine-tuning (SFT) and reinforcement learning (RL), as a hybrid training baseline to validate the generality of our adversarial loss. All experiments were based on MS-Swift Zhao et al. (2025),VeRL Sheng et al. (2025) and Trinity-Rft Pan et al. (2025). Detailed experimental settings can be found in Appendix B.1.

## 4.2 MAIN RESULT

**Objective Metrics and Experimental Analysis.** Our primary experimental results focus on two training paradigms: staged training and unified training with the UniAPL adversarial loss. Figure 2 presents the staged UniAPL results and compares them against the Qwen3 series models on instruction-following tasks. For MultiIF, we evaluate only the single-turn multilingual instruction-following ability. The full set of evaluation metrics is provided in B.4. As shown in Table 2 and Table 3, several key findings emerge from the staged training setup. First, in the offline supervised fine-tuning (SFT) phase, A-SFT slightly mitigates the degradation in general capabilities and achieves larger gains on instruction-following benchmarks compared to standard SFT. Second, during the online training phase, we observe that A-GRPO surprisingly achieves performance comparable to the two-stage SFT→GRPO approach on task-specific metrics. This is likely because A-GRPO continuously guides the student policy to align with the teacher policy during training, effectively serving as an implicit SFT warm-up. This effect is particularly evident in the 4B model, where A-GRPO directly prevents the degradation of general capabilities typically introduced by SFT. Furthermore, in both our two-stage and unified training experiments, the UniAPL adversarial loss still leads to improved performance. Moreover, by enhancing the model's instruction-following capability, the adversarial loss enables deeper comprehension of instructions, leading to superior performance across various capabilities compared to the base model.

| Method | Benchmarks | | | | | |
|---|---|---|---|---|---|---|
| | Avg | I-Following | English | Coding | Mathematics | Chinese |
| *Qwen3-0.6B* | 40.81 | 56.62 | 32.61 | 32.63 | 42.29 | 43.25 |
| *SFT* | 38.48 | 59.97 | 25.20 | 29.43 | 39.56 | 44.33 |
| *GRPO* | 40.93 | 74.09 | 32.39 | 17.58 | 41.49 | 43.11 |
| *SFT→GRPO* | 42.56 | 77.40 | 35.09 | 21.60 | 39.57 | 44.38 |
| *SFT→A-GRPO* | **43.61** | 79.57 | 32.71 | 29.00 | 39.95 | 44.10 |
| *A-SFT* | 39.83 | 60.72 | 26.67 | **33.93** | 40.14 | 44.10 |
| *A-GRPO* | 41.99 | 79.58 | 31.36 | 19.79 | 41.04 | 43.96 |
| *A-SFT→A-GRPO* | 43.31 | **79.86** | **36.48** | 33.72 | **43.72** | **44.47** |
| *Qwen3-4B* | 62.52 | 80.10 | 45.42 | 70.31 | 59.47 | 67.36 |
| *SFT* | 60.80 | 80.60 | 41.13 | 68.62 | 56.92 | 68.51 |
| *GRPO* | 68.11 | 86.90 | 53.42 | 71.43 | **66.52** | 70.42 |
| *SFT→GRPO* | 65.75 | 89.75 | 49.04 | 71.54 | 59.40 | 70.53 |
| *SFT→A-GRPO* | 68.11 | 90.45 | 48.25 | 72.06 | 60.53 | 70.93 |
| *A-SFT* | 60.67 | 82.47 | 38.78 | 70.34 | 56.65 | 68.07 |
| *A-GRPO* | **68.39** | 88.17 | **54.29** | 73.15 | 65.12 | 69.89 |
| *A-SFT→A-GRPO* | 67.93 | **90.65** | 51.16 | **73.26** | 63.65 | **71.45** |

Table 2: Comparative results of different training methods across multiple benchmarks (IFEval, MulitIF, MMLU, MMLU-Pro, GPQA-Diamond, HumanEval, MBPP, GSM8K, MATH-500, TheoremQA, CMMLU, CEval) for models with `Qwen3-0.6B` and `Qwen3-4B`.

| Method | Benchmarks | | | | | |
|---|---|---|---|---|---|---|
| | Avg | I-Following | English | Coding | Mathematics | Chinese |
| *Qwen3-0.6B* | 40.81 | 56.62 | 32.61 | 32.63 | **42.29** | 43.25 |
| *CHORD* | 41.19 | 73.52 | 29.92 | 26.09 | 38.52 | 44.90 |
| *A-CHORD* | **42.09** | **75.25** | **32.74** | 23.89 | 39.45 | **45.14** |
| *Qwen3-4B* | 62.52 | 80.10 | 45.42 | 70.31 | 59.47 | 67.36 |
| *CHORD* | **69.30** | 86.97 | 54.86 | **73.76** | 67.67 | 71.30 |
| *A-CHORD* | 69.27 | **88.27** | 55.46 | 70.32 | **68.14** | 71.66 |

Table 3: Comparative results of unified training methods across multiple benchmarks for models with `Qwen3-0.6B` and `Qwen3-4B`.

**Theoretical Validation.** To verify that UniAPL effectively learns from the teacher model while mitigating policy drift, we re-evaluate models trained with SFT, GRPO, and UniAPL on the 38K instances from IFBench. We report the distribution of response lengths in Figure 3. Our results show that SFT captures only the surface-level length pattern, while GRPO relies on self-exploration. UniAPL, by emulating the teacher's reasoning, produces responses whose length distribution aligns more closely with the teacher than GRPO does. An example can be found in Appendix B.6. In addition, to verify that UniAPL continuously constrains the student model to learn from the teacher, we sample 500 responses from the dataset and compute the total log-likelihood (logp) under three models: the teacher, the student trained with GRPO, and the student trained with UniAPL. The results are summarized in Figure 4. The narrower distribution and reduced divergence under UniAPL indicate closer alignment between student and teacher outputs, demonstrating its superior policy regularization and more effective knowledge transfer compared to GRPO.

## 4.3 ABLATION STUDY

**Ablation on UniAPL Adversarial Loss Method.** We validate the effectiveness of the base model and ground-truth (GT) Dataset in the appendix B.1. Furthermore, we analyze the impact of the UniAPL discriminator coefficients in conjunction with GT references and the KL divergence loss

| Method | Benchmarks | | | | | |
|---|---|---|---|---|---|---|
| | Avg | I-Following | English | Coding | Mathematics | Chinese |
| *RLVR* | 47.53 | 76.25 | **36.43** | 36.29 | 43.83 | 52.22 |
| *RLVR+POLAR* | 46.70 | 68.47 | 36.01 | 36.80 | **46.06** | 51.82 |
| *RLVR+$A_{\text{coef}}$* | 47.68 | 73.09 | 35.96 | **38.94** | 45.09 | **52.51** |
| *w $A_{\text{coef}}$* | **47.83** | **78.94** | 36.23 | 35.57 | 43.91 | 52.26 |

Table 4: Table 4: RLVR denotes the setup that uses only verifiable rewards, corresponding to the standard SFT→ GRPO pipeline. The remaining three variants correspond to our three ablation configurations.

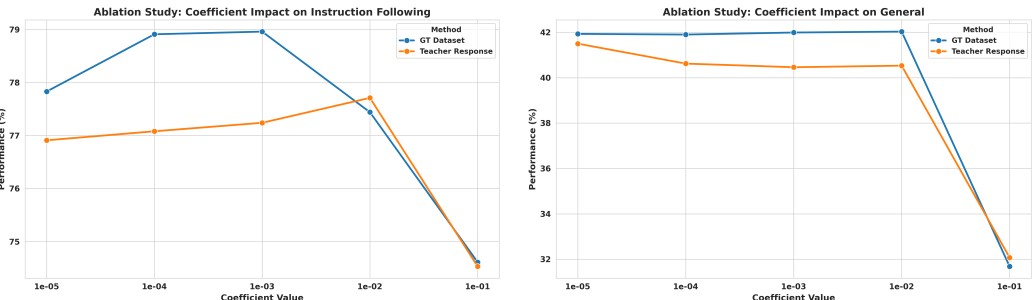

Figure 5: Ablation study of discriminator coefficients in UniAPL on instruction-following tasks.

Figure 6: Ablation study of discriminator coefficients in UniAPL on general-purpose performance.

in Appendix B.2. In addition, we conduct ablation studies on reward-model (RM) scoring tasks and mathematical reasoning tasks to further demonstrate the broad applicability of our approach.To investigate the contribution of the UniAPL adversarial loss,we conducted a series of ablation experiments under the SFT→AGRPO training paradigm using the base model and GT Dataset. Specifically, we compared three settings in Table 4, where the base method employs RLVR (Reinforcement Learning with Verifiable Rewards)'s GRPO. (i) directly adding the raw output of POLAR to the verifiable reward, (ii) adding the transformed output to the verifiable reward, (iii) using the transformed output as a separate adversarial loss term. Among these settings, using it as an independent adversarial loss achieved the best performance,which aligns with our theoretical expectation.

**Ablation on the UniAPL Adversarial Loss Coefficient $\lambda_{\text{adv}}$.** We conducted extensive ablation studies on the UniAPL adversarial loss coefficient $\lambda_{\text{adv}}$, varying it from 0.1 to $1 \times 10^{-5}$, which spans the typical magnitude difference relative to the policy gradient loss. In all experiments, the coefficient of the KL loss was fixed at 0.001. Figure 5and 6 presents all coefficient settings evaluated on both GT-based data and teacher-generated responses.We suggest setting the coefficient within the range of 0.01 to 0.0001, which is approximately one order of magnitude lower than the policy gradient loss, serving as an auxiliary signal. A value that is too high may dominate the optimization with the adversarial term and suppress the model's exploration, while a value that is too low may lead to insufficient performance gains.

## 5 CONCLUSION

This work introduces Unified Adversarial Preference Learning (UniAPL), a framework that mitigates the distributional mismatch in sequential SFT-then-RL alignment by reframing alignment as a single-stage constrained optimization problem. UniAPL employs an adversarial objective that enforces distributional consistency, enabling dense SFT signals to dynamically regularize RL exploration. This joint process anchors the policy to expert behavior while allowing preference learning to extend beyond the static expert distribution. Although primarily methodological, UniAPL delivers notable empirical gains and simplifies the training pipeline, supporting our hypothesis that unifying imitative and preference-based learning is a more effective and efficient alignment strategy. We expect UniAPL to serve as a practical foundation for integrating richer supervision and developing more robust approaches for steering advanced AI systems.

## ETHICS STATEMENT

This work does not involve human subjects, private or sensitive data, or any personally identifiable information. All datasets employed in our experiments are publicly available and have been used in accordance with their respective licenses. We acknowledge the potential societal risks associated with large language models, including issues of fairness, bias, and misuse. While these concerns are not the primary focus of this work, we have taken steps to ensure responsible experimentation, including transparent reporting of datasets, models, and evaluation protocols. The release of our code and models will be accompanied by appropriate documentation and usage guidelines to mitigate unintended applications.

## REPRODUCIBILITY STATEMENT

We have made every effort to ensure that our results are reproducible. The main text provides comprehensive descriptions of the proposed methodology, model architecture, training objectives, and evaluation protocols. Additional implementation details, including hyperparameter settings and ablation configurations, are documented in the appendix. All datasets employed in our experiments are publicly accessible, and the complete data preprocessing pipeline is described in the supplementary materials. To further facilitate independent verification and extension of our work, we will release the source code, trained model checkpoints, and experiment scripts in the near future.

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

# A  THEORETICAL AND IMPLEMENTATION DETAILS OF THE UNIAPL FRAMEWORK

This appendix provides a deeper exploration of the theoretical underpinnings and implementation details of our proposed framework.

## A.1  THE UNIFIED VIEW: SFT AS A SPECIAL CASE OF PREFERENCE LEARNING

Our framework is built on the premise that SFT and RL are not disparate stages but points on a continuum of preference learning. To formalize this, we re-interpret SFT as an extreme and limiting case of preference learning.

**The Conventional View: SFT as Imitation Learning.**  Traditionally, SFT is understood as imitation learning. The objective is to train a policy $\pi_\theta$ to mimic an expert by minimizing the Negative Log-Likelihood (NLL) of a single ground-truth response $y^*$:

$$\mathcal{L}_{\text{SFT}}(\theta) = -\log \pi_\theta(y^*|x)$$

Minimizing this loss is equivalent to maximizing the probability $\pi_\theta(y^*|x)$. In this view, SFT is about replicating a single correct answer, distinct from preference learning, which involves choosing between multiple answers.

**The Unified View: SFT as Preference Learning with a Dirac Delta Reward.**  Our framework unifies these concepts by redefining "preference" for the SFT case. While standard preference learning (e.g., DPO) is relative (a "winner" $y_w$ is preferred over a "loser" $y_l$), SFT can be seen as expressing an absolute and infinite preference.

Specifically, the ground-truth response $y^*$ is **infinitely preferred** over any other possible response $y \neq y^*$. There is no concept of a response being "close" or "better" than another incorrect response. To model this, we define a reward function using the Dirac delta function, $\delta(\cdot)$:

$$R(y|x) = \delta(y - y^*)$$

This function grants an infinite reward if the policy generates the exact ground-truth response and zero reward for any deviation.

**Mathematical Equivalence of Reward Maximization and NLL Minimization.**  We can formally show that the standard SFT objective is a special case of the general reward-maximization objective.

1. The standard reinforcement learning objective is to maximize the expected reward for a response $y$ sampled from the policy $\pi_\theta(\cdot|x)$:

$$\max_\theta \mathbb{E}_{y \sim \pi_\theta(\cdot|x)}[R(y|x)]$$

2. Substituting our Dirac delta reward function:

$$\max_\theta \mathbb{E}_{y \sim \pi_\theta(\cdot|x)}[\delta(y - y^*)]$$

3. The expectation is an integral (or sum) of the reward of each possible output $y$ weighted by its probability $\pi_\theta(y|x)$. Due to the properties of the Dirac delta function, the integral is non-zero only at the single point $y = y^*$. Therefore, the expectation collapses to be proportional to the probability of generating $y^*$:

$$\mathbb{E}_{y \sim \pi_\theta(\cdot|x)}[\delta(y - y^*)] \propto \pi_\theta(y^*|x)$$

4. To maximize this expectation, one must maximize the probability term:

$$\max_\theta \pi_\theta(y^*|x)$$

5. Since the logarithm is a monotonic function, this is equivalent to maximizing its logarithm:

$$\max_\theta \log \pi_\theta(y^*|x)$$

6. Finally, maximizing a function is equivalent to minimizing its negation:

$$\min_{\theta}(-\log \pi_\theta(y^*|x))$$

This final expression is precisely the NLL loss for SFT. We have formally shown that the objective of SFT is a special case of the general reward-maximization objective, where the reward function is a Dirac delta centered on the expert demonstration.

## A.2 THEORETICAL SIGNIFICANCE OF THE UNIFIED VIEW

This unification is critically important for several reasons:

1. **Establishes a Coherent Theoretical Framework.** It places SFT, DPO, and online RL under a single, elegant objective function, $J(\theta)$. This provides a cohesive narrative for the entire post-training pipeline, viewing it not as a sequence of disparate steps but as a unified process.

2. **Reveals a Curriculum of Reward Signals.** It frames post-training as a principled progression where the nature of the reward signal evolves. Stage 1 (SFT) uses a sharp, sparse Dirac delta signal to instill absolute knowledge. Subsequent stages (DPO/GRPO) replace this sharp signal with a smooth preference landscape, teaching the model to navigate nuanced trade-offs and generalize in a continuous space of response quality.

## A.3 IMPLEMENTATION OF THE ADVERSARIAL DISCRIMINATOR

As detailed in the main text, our adversarial objective $\mathcal{L}_{\text{ADV}}$ relies on a discriminator $D_\phi$. To implement this, we adapt the methodology from POLAR, a model designed to measure the distance between generation strategies. We use a pre-trained POLAR model as the backbone for our discriminator. The raw distance score from POLAR is then transformed into a well-behaved adversarial loss coefficient for our objective. The full procedure is detailed in Algorithm 1.

---

**Algorithm 1** Adversarial Loss Coefficient Computation Based on POLAR Distance

Prompt $x$, Teacher policy $\pi_t$, Student policy $\pi_s$ Adversarial loss coefficient $coef \in [-1, 1]$ Obtain teacher response $y_t = \pi_t(x)$ and student response $y_s = \pi_s(x)$ Feed $(x, y_t, y_s)$ into POLAR model to get BT distance: $r = \text{POLAR}(x, y_t, y_s)$ Normalize distance $r$ into similarity score $p$ using sigmoid:

$$p = \sigma(r) = \frac{1}{1 + e^{-r}} \quad (p \in (0, 1))$$

Convert similarity score $p$ into an adversarial loss coefficient $coef$:

$$coef = 1 - 8(p - 0.5)^2 \quad (coef \in [-1, 1])$$

**return** $coef$

---

The final $coef$ serves as the output of our discriminator, $D_\phi(x, y_t, y_s)$, which is then used to compute $\mathcal{L}_{\text{ADV}}$.

## A.4 ALTERNATIVE PREFERENCE OPTIMIZATION FORMULATIONS

Our UniAPL framework is compatible with a wide range of preference optimization algorithms. While the main text focuses on a policy-gradient approach (GRPO), here we detail prominent reward-model-free alternatives.

### A.4.1 DIRECT PREFERENCE OPTIMIZATION (DPO)

DPO is a widely-used algorithm that formulates preference learning as a direct classification problem on human preferences.

**Data Format.** The preference dataset $D_{\text{PREF}}$ for DPO consists of tuples $(x, y_w, y_l)$, where for a given prompt $x$, $y_w$ is the "winner" (preferred) response and $y_l$ is the "loser" (dispreferred) response.

**Loss Function.** The DPO loss is formulated to directly increase the likelihood of the winner response while decreasing the likelihood of the loser response, relative to a fixed reference policy $\pi_{\text{ref}}$. It is defined as:

$$\mathcal{L}_{\text{DPO}}(\theta, \pi_{\text{ref}}) = -\mathbb{E}_{(x, y_w, y_l) \sim D_{\text{PREF}}} \left[ \log \sigma \left( \beta \log \frac{\pi_\theta(y_w|x)}{\pi_{\text{ref}}(y_w|x)} - \beta \log \frac{\pi_\theta(y_l|x)}{\pi_{\text{ref}}(y_l|x)} \right) \right]$$

Here, $\beta$ is a temperature parameter that controls the strength of the preference modeling, and $\sigma$ is the sigmoid function.

### A.4.2 KAHNEMAN-TVERSKY OPTIMIZATION (KTO)

KTO offers a different perspective by removing the need for pairwise comparisons altogether. It instead relies on binary labels for individual examples.

**Data Format.** The preference dataset $D_{\text{PREF}}$ for KTO consists of tuples $(x, y, l)$, where $l \in \{\text{desirable}, \text{undesirable}\}$ is a binary label indicating whether the response $y$ to prompt $x$ is good or bad.

**Objective.** The KTO loss is designed based on principles from human prospect theory. It has two main components: one term encourages the policy to increase the likelihood of "desirable" examples, and another, more heavily weighted term, strongly discourages the policy from generating "undesirable" examples. This asymmetry reflects the human tendency to be more sensitive to losses than to equivalent gains.

## B DETAILED BENCHMARK RESULTS AND EXAMPLE

### B.1 DETAILED EXPERIMENT TRAINING SETTING

**General Setting.** All experiments were conducted on eight H100 GPUs. The learning rate was set to $1 \times 10^{-5}$ for the SFT stage, $1 \times 10^{-7}$ for the GRPO stage, and $1 \times 10^{-6}$ for the mixed training, following the default configuration of CHORD. After completing one epoch of SFT, we perform one epoch of RL. Under our unified training framework, the same SFT and RL datasets are used. RL is trained with 8 rollouts, and the ratio between SFT and RL samples at each training step is fixed at 1:4. Consequently, the SFT stage runs for 2 epochs in total, while the RL stage corresponds to 1 epoch. We did not apply early stopping, as doing so would compromise the fairness and comparability of the experiments.

**Chat model setting** Since the Qwen3 models require manual control of the *think* process, all GRPO experiments based on the Chat models were performed with `enable_thinking=False`. Each model response begins with the prefix `<think>\n\n</think>\n\n`. Similarly, for SFT training on Chat models, this prefix was prepended to the response part of the data, and the loss on empty think segments was excluded from the computation.

**Base model Setting** To validate the effectiveness of our approach on the base model and ground-truth references, we conducted several ablation studies on the Qwen3-Base model using the reference answers. To eliminate the potential interference of reasoning-related labels, we applied supervised fine-tuning (SFT) with the `Qwen2.5` chat template, and removed the `<think>\n\n</think>\n\n` prefix from the SFT response data. Since A-SFT requires an additional rollout step and the base model lacks inherent conversational ability, experiments based on the base model were carried out under the SFT→GRPO training paradigm. As shown in Table 5, when using the ground-truth (GT) reference responses from the dataset as evaluation targets, initializing from the base model achieves better generalization performance than initializing from a chat model, while reaching comparable instruction-following capability.

| Method | Benchmarks | | | | | |
|---|---|---|---|---|---|---|
| | Avg | I-Following | English | Coding | Mathematics | Chinese |
| *0.6B-Basenothink/GT* | - | - | - | - | - | - |
| *SFT* | 43.97 | 60.13 | 33.32 | 35.87 | 43.60 | **52.43** |
| *SFT→ GRPO* | 47.53 | 76.25 | **36.43** | **36.29** | 43.83 | 52.22 |
| *SFT→ A-GRPO* | **47.83** | **78.96** | 36.23 | 35.57 | **43.91** | 52.26 |

Table 5: Experimental results of base model on the ground-truth (GT) dataset.SFT→GRPO corresponds to RLVR in Table 16, while SFT and SFT→A-GRPO corresponds to the entry with rate 0.001 in Table 14.

## B.2 SUPPLEMENTARY ABLATION ANALYSIS

**Ablation on UniAPL discriminator coefficients with GT References and KL Loss** We further conducted an ablation study on the UniAPL discriminator coefficient using ground-truth (GT) reference answers. Specifically, the GT answers were treated as the target, and both the teacher and student outputs were passed through POLAR to obtain their respective scores, $r_t$ and $r_s$. In the policy space, this corresponds to the teacher and student policies lying within a circle centered at the reference policy with radius $\max(r_t, r_s)$. The difference between these scores, $\Delta r$, approximates the distance $R$ between the two policies. Subsequently, the distances between these scores were calculated according to the algorithm described in the Appendix A.3 and used to derive the discriminator coefficient. Table 6 presents the results of this ablation study. Finally, we conducted ablation experiments on the KL loss to examine its effect on training stability and performance. When the KL loss was disabled, the model's policy optimization was guided solely by the PG loss and adversarial loss. Through this experimentthe presence or absence of reference answers in the calculation of the discriminator coefficient has little impact on the final performance. Disabling the KL loss allows the model to achieve better performance on this specific task, but it may leads to a collapse of its general capabilities on other tasks.

| Method | Benchmarks | | | | | |
|---|---|---|---|---|---|---|
| | Avg | I-Following | English | Coding | Mathematics | Chinese |
| *Directly Distinguish* | - | - | - | - | - | - |
| *Use KL* | 45.86 | 77.24 | 33.29 | **37.41** | 38.84 | **52.31** |
| *NO KL* | **47.52** | **80.65** | **36.03** | 35.39 | **41.90** | 52.21 |
| *With GT Response* | - | - | - | - | - | - |
| *Use KL* | **46.25** | 77.43 | 32.48 | 36.40 | 41.98 | **51.95** |
| *NO KL* | 40.65 | **80.38** | **36.62** | **38.01** | 12.64 | 51.62 |

Table 6: Ablation study of UniAPL discriminator coefficients with ground-truth references and KL loss.Detailed experimental results are provided in Table 16

**Ablation on UniAPL Discriminator and Teacher** To validate our theory, we conducted ablation experiments on the discriminator and teacher responses to avoid potential biases caused by using a single discriminator or teacher. For our instruction fine-tuning task, the discriminator selection experiment was performed under the SFT→A-GRPO paradigm. We chose the `BGE-M3` model, which is designed for retrieving text similarity, as the discriminator. The student model remained `Qwen3-0.6B-Base`, while for the teacher model, we selected a smaller model, `Qwen3-32B`. Table 7 demonstrates the effectiveness of our theory: as long as the discriminator benefits the task, it has a positive effect. Combined with the results in Table 5, the quality of teacher responses does not significantly alter the overall outcome. The model primarily explores on its own, while the adversarial signal serves as a guiding influence.

**Ablation on Different Task** To validate the broad effectiveness of our theory, we conducted experiments on both RLHF tasks involving reward model scoring and on mathematical tasks. For the reward model task, we used the same instruction data but removed the rule-based rewards, and employed the `internlm2-20b-Reward` model to score model responses. As shown in Table 8,

| Method | Benchmarks | | | | | |
|---|---|---|---|---|---|---|
| | Avg | I-Following | English | Coding | Mathematics | Chinese |
| *Discriminator Ablation* | - | - | - | - | - | - |
| *BGE-M3* | **46.00** | 77.07 | **34.94** | 34.57 | **39.92** | 52.11 |
| *POLAR* | 45.86 | **77.24** | 33.29 | **37.41** | 38.84 | **52.31** |
| *Teacher Ablation* | - | - | - | - | - | - |
| *Qwen3-235B-Instruct* | 45.86 | **77.24** | 33.29 | **37.41** | 38.84 | 52.31 |
| *Qwen3-32B* | **46.17** | 76.32 | **35.15** | 36.70 | **39.26** | **52.38** |

Table 7: Ablation study of UniAPL discriminator and teacher responses.Detailed experimental results are provided in Table 17

we report model performance at 1/3 and 2/3 of an epoch, since reward models are prone to reward hacking. Although the reward scores continue to increase during training, the performance of `Qwen3-0.6B-Chat` completely collapses when trained on 150K instruction data. However, with the addition of adversarial gradients, the degradation in instruction-following ability is mitigated.

For the mathematical tasks, we employed the CHORD method and sampled 20K examples from OpenR1-Math-220k, and the same subset was used for experiments with the Luffy to ensure a fair comparison. To ensure model consistency, the ablation experiments for the mathematical tasks were conducted on `Qwen2.5-7B-Instruct`. Table 9 presents the results after training for one epoch, where A-CHORD achieves the best performance. We did not train for multiple epochs, so the metrics may not fully align with those reported in their paper.

| Method | Benchmarks | | | | | |
|---|---|---|---|---|---|---|
| | Avg | I-Following | English | Coding | Mathematics | Chinese |
| *Internlm2-20b-Reward* | - | - | - | - | - | - |
| *RM-1/3* | 27.14 | 42.03 | 19.62 | 28.58 | 25.39 | 24.71 |
| *RM-2/3* | 4.54 | 16.65 | 1.25 | 0.92 | 0.50 | 7.05 |
| *A-RM-1/3* | 26.90 | 43.81 | 12.74 | 31.50 | 26.58 | 27.13 |
| *A-RM-2/3* | 6.44 | 19.62 | 1.44 | 0.31 | 4.63 | 9.59 |

Table 8: Ablation study on the reward model task. Although this task is prone to reward hacking, the addition of adversarial gradients mitigates such behavior.Detailed experimental results are provided in Table 17

| Method | Benchmarks | | | |
|---|---|---|---|---|
| | Avg | AIME24 | AIME25 | AMC |
| *Qwen2.5-7B-Instruct* | 20.72 | 11.7 | 6.66 | 43.80 |
| *Luffy* | 23.69 | 10.00 | 14.16 | **46.92** |
| *CHORD* | 24.25 | **16.25** | 15.00 | 41.51 |
| *A-CHORD* | **24.64** | 15.42 | **16.25** | 42.26 |

Table 9: Ablation study on the Math task. Evaluation was conducted on the AIME24, AIME25, and AMC datasets.

### B.3 STEP TIME AND REWARD ANALYSIS

According to our theory, the additional overhead we introduce mainly arises from two sources. However, this overhead is negligible in a reward-model-scoring process, which aligns with our Occam's razor-inspired idea of avoiding real-time updates.

The first source is the addition of losses. Here, we primarily introduce a scalar multiplied by the log-likelihood, which incurs almost no extra cost during actor updates. We provide a comparison of actor update times during training in Figure 8, showing that the time remains essentially unchanged.

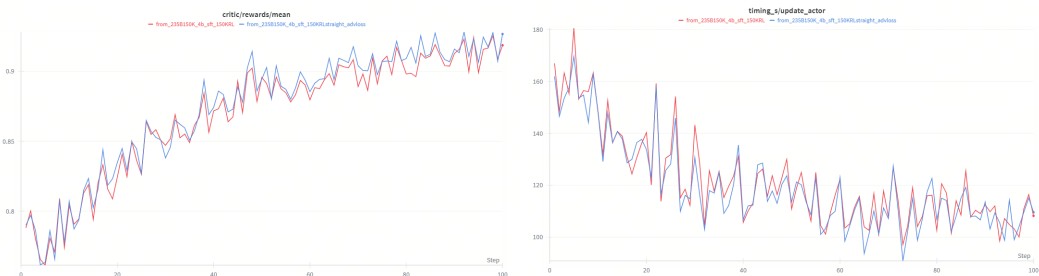

Figure 7: Comparison of reward acquisition during the first 100 training steps for the 4B model. The blue curve denotes the adversarial-enhanced variant, while the red curve corresponds to the GRPO baseline.

Figure 8: Comparison of actor update time in the first 100 steps of training for the 4B model.The blue curve denotes the adversarial-enhanced variant, while the red curve corresponds to the GRPO baseline.

The second source is the discriminator evaluation. This is typically deployed as a network service. In the RLHF/RLAIF process, the discriminator is usually a relatively small model and is much faster than the reward model; using multithreading to call the API introduces almost no additional overhead. In the RLVR process, however, the discriminator introduces extra cost depending on its inference speed. A-SFT requires a model rollout, which is time-consuming; therefore, we do not recommend using A-SFT directly, as the gains are limited. Instead, we integrate it into A-GRPO and unified training, which is one of the motivations for introducing our unified framework.

Furthermore, we report the reward obtained in the first 100 steps with adversarial gradients for the 4B model, corresponding to 1/3 of the epoch, as shown in Figure 7. Training with adversarial gradients results in higher reward acquisition.

## B.4 DETAILED BENCHMARK RESULTS

| Benchmark | UniAPL Model | | Qwen3 Model | | | | | | |
|---|---|---|---|---|---|---|---|---|---|
| | 0.6B | 4B | 0.6B | 1.7B | 4B | 4B-2507 | 8B | 32B | 235B-2507 |
| Avg | 43.31 | 67.93 | 40.81 | 52.83 | 62.52 | 65.34 | 72.43 | 67.19 | 77.05 |
| **AvgIF(Report)** | **79.86** | **90.65** | **56.62** | **67.59** | **80.10** | **82.84** | **84.48** | **85.04** | **89.18** |
| IFEVAL | 82.99 | 90.02 | 55.64 | 66.73 | 78.19 | 80.59 | 82.26 | 84.29 | 87.99 |
| MultiIF | 76.73 | 91.28 | 57.60 | 68.44 | 82.00 | 85.09 | 86.69 | 85.79 | 90.37 |
| Mmlu | 45.28 | 69.86 | 43.91 | 59.05 | 69.90 | 75.89 | 82.78 | 73.89 | 88.39 |
| Mmlu-pro | 27.08 | 56.35 | 23.63 | 20.53 | 35.56 | 33.10 | 63.20 | 20.13 | 39.45 |
| GPQA | 28.79 | 27.27 | 30.30 | 24.24 | 30.81 | 25.76 | 39.90 | 45.45 | 59.60 |
| Humaneval | 39.63 | 81.71 | 41.46 | 67.07 | 75.61 | 83.54 | 89.63 | 87.20 | 96.95 |
| Mbpp | 27.80 | 64.80 | 23.80 | 51.80 | 65.00 | 69.20 | 77.80 | 69.60 | 84.60 |
| Gsm8k | 42.91 | 90.14 | 61.18 | 74.37 | 83.78 | 85.14 | 84.46 | 88.17 | 84.99 |
| Math-500 | 44.62 | 65.92 | 49.06 | 63.24 | 70.26 | 70.98 | 73.88 | 91.36 | 93.50 |
| TheoremQA | 15.88 | 34.88 | 16.62 | 20.50 | 24.38 | 21.25 | 25.00 | 14.75 | 18.12 |
| Cmmlu | 46.60 | 72.45 | 45.36 | 59.96 | 72.09 | 76.76 | 84.87 | 73.64 | 89.44 |
| Ceval | 41.38 | 70.45 | 41.14 | 58.02 | 62.62 | 76.76 | 78.70 | 72.02 | 91.21 |

Table 10: Detailed results of the unified experiments

| Benchmark | 0.6B | SFT | GRPO | S→G | S→A-G | A-SFT | A-GRPO | A-S→A-G |
|---|---|---|---|---|---|---|---|---|
| Avg | 40.81 | 38.48 | 40.93 | 42.56 | 43.61 | 39.83 | 41.99 | 43.31 |
| IFEVAL | 55.64 | 60.26 | 74.68 | 80.78 | 82.07 | 61.74 | 82.26 | 82.99 |
| MultiIF | 57.60 | 59.68 | 73.50 | 74.01 | 77.06 | 59.70 | 76.89 | 76.73 |
| Mmlu | 43.91 | 46.16 | 42.01 | 43.79 | 43.35 | 47.07 | 40.54 | 45.28 |
| Mmlu-pro | 23.63 | 21.37 | 26.36 | 24.61 | 25.99 | 23.34 | 25.77 | 27.08 |
| GPQA | 30.30 | 8.08 | 28.79 | 36.87 | 28.79 | 9.60 | 27.78 | 28.79 |
| Humaneval | 41.46 | 40.85 | 34.76 | 37.20 | 37.20 | 41.46 | 35.98 | 39.63 |
| Mbpp | 23.80 | 18.00 | 0.40 | 6.00 | 20.80 | 26.40 | 3.60 | 27.80 |
| Gsm8k | 61.18 | 59.21 | 58.38 | 62.02 | 61.26 | 58.07 | 60.27 | 42.91 |
| Math-500 | 49.06 | 43.16 | 46.96 | 43.68 | 43.00 | 44.84 | 48.48 | 44.62 |
| TheoremQA | 16.62 | 16.32 | 19.12 | 13.00 | 15.38 | 17.50 | 14.37 | 15.88 |
| Cmmlu | 45.36 | 45.99 | 45.00 | 45.88 | 46.01 | 46.44 | 44.84 | 46.60 |
| Ceval | 41.14 | 42.66 | 41.22 | 42.88 | 42.18 | 41.76 | 43.08 | 41.38 |

Table 11: Detailed staged experimental results on Qwen3-0.6B

| Benchmark | 4B | SFT | GRPO | S→G | S→A-G | A-SFT | A-GRPO | A-S→A-G |
|---|---|---|---|---|---|---|---|---|
| Avg | 62.52 | 60.80 | 68.11 | 65.75 | 66.10 | 60.67 | 68.39 | 67.93 |
| IFEVAL | 78.19 | 77.82 | 87.06 | 88.91 | 90.02 | 81.15 | 88.54 | 90.02 |
| MultiIF | 82.00 | 83.37 | 86.74 | 90.58 | 90.88 | 83.79 | 87.80 | 91.28 |
| Mmlu | 69.90 | 72.78 | 70.02 | 71.47 | 70.96 | 71.91 | 70.00 | 69.86 |
| Mmlu-pro | 35.56 | 40.51 | 54.88 | 53.43 | 54.59 | 31.29 | 51.46 | 56.35 |
| GPQA | 30.81 | 10.10 | 35.35 | 22.22 | 19.19 | 13.13 | 41.41 | 27.27 |
| Humaneval | 75.61 | 76.83 | 78.66 | 79.88 | 82.32 | 79.27 | 81.10 | 81.71 |
| Mbpp | 65.00 | 60.40 | 64.20 | 63.20 | 61.80 | 61.40 | 65.20 | 64.80 |
| Gsm8k | 83.78 | 86.81 | 89.92 | 89.08 | 89.31 | 86.73 | 87.04 | 90.14 |
| Math-500 | 70.26 | 62.20 | 70.14 | 63.36 | 62.66 | 63.72 | 70.7 | 65.92 |
| TheoremQA | 24.38 | 21.75 | 39.50 | 25.75 | 29.62 | 19.50 | 37.62 | 34.88 |
| Cmmlu | 72.09 | 72.18 | 72.03 | 72.48 | 72.63 | 71.80 | 72.07 | 72.45 |
| Ceval | 62.62 | 64.84 | 68.80 | 68.58 | 69.22 | 64.34 | 67.70 | 70.45 |

Table 12: Detailed staged experimental results on Qwen3-4B

| Benchmark | 0.6B Model | | | 4B Model | | |
|---|---|---|---|---|---|---|
| | SFT | CHORD | ACHORD | SFT | CHORD | ACHORD |
| Avg | 40.81 | 41.19 | 42.09 | 62.52 | 69.30 | 69.27 |
| IFEVAL | 55.64 | 74.49 | 76.16 | 78.19 | 87.06 | 87.80 |
| MultiIF | 57.60 | 72.54 | 74.34 | 82.00 | 86.87 | 88.74 |
| Mmlu | 43.91 | 41.49 | 44.10 | 69.90 | 70.29 | 69.94 |
| Mmlu-pro | 23.63 | 25.53 | 25.33 | 35.56 | 55.90 | 56.04 |
| GPQA | 30.30 | 22.73 | 28.79 | 30.81 | 38.38 | 40.40 |
| Humaneval | 41.46 | 35.98 | 35.98 | 75.61 | 81.71 | 77.44 |
| Mbpp | 23.80 | 16.20 | 16.20 | 65.00 | 65.80 | 63.20 |
| Gsm8k | 61.18 | 55.57 | 55.57 | 83.78 | 87.41 | 90.86 |
| Math-500 | 49.06 | 43.86 | 43.86 | 70.26 | 68.34 | 72.30 |
| TheoremQA | 16.62 | 16.12 | 16.12 | 24.38 | 47.25 | 41.25 |
| Cmmlu | 45.36 | 45.85 | 45.85 | 72.09 | 71.96 | 72.69 |
| Ceval | 41.14 | 43.94 | 43.94 | 62.62 | 70.63 | 70.62 |

Table 13: Detailed results of the unified experiments

| Benchmark | SFT | 0.1 | 0.01 | 0.001 | 0.0001 | 0.00001 |
|---|---|---|---|---|---|---|
| Avg | 43.97 | 38.18 | 47.53 | 47.83 | 47.67 | 47.61 |
| IFEVAL | 62.66 | 76.52 | 78.74 | 80.78 | 81.33 | 79.30 |
| MultiIF | 57.59 | 72.70 | 76.13 | 77.14 | 76.49 | 76.36 |
| Mmlu | 52.90 | 50.92 | 51.99 | 52.02 | 51.91 | 52.56 |
| Mmlu-pro | 27.88 | 24.89 | 28.19 | 27.38 | 27.42 | 26.72 |
| GPQA | 19.19 | 25.76 | 30.30 | 29.29 | 25.25 | 28.28 |
| Humaneval | 32.93 | 3.66 | 35.37 | 32.93 | 34.15 | 33.54 |
| Mbpp | 38.80 | 35.80 | 38.20 | 38.20 | 38.60 | 37.00 |
| Gsm8k | 62.70 | 26.99 | 62.77 | 63.38 | 65.81 | 64.75 |
| Math-500 | 45.10 | 16.44 | 42.22 | 43.46 | 44.00 | 44.54 |
| TheoremQA | 23.00 | 21.61 | 22.50 | 24.88 | 22.75 | 23.75 |
| Cmmlu | 51.02 | 49.94 | 50.13 | 50.17 | 50.15 | 50.15 |
| Ceval | 53.84 | 52.94 | 53.87 | 54.34 | 54.21 | 54.21 |

Table 14: Detailed experimental results on GT coef

| Benchmark | SFT | 0.1 | 0.01 | 0.001 | 0.0001 | 0.00001 |
|---|---|---|---|---|---|---|
| Avg | 42.47 | 38.09 | 46.21 | 45.86 | 46.22 | 46.89 |
| IFEVAL | 57.12 | 75.97 | 78.93 | 78.74 | 78.74 | 78.19 |
| MultiIF | 59.53 | 73.09 | 76.49 | 75.70 | 75.42 | 75.62 |
| Mmlu | 52.82 | 49.81 | 52.82 | 53.09 | 53.17 | 53.47 |
| Mmlu-pro | 27.04 | 23.23 | 26.81 | 27.08 | 26.85 | 28.12 |
| GPQA | 10.10 | 28.28 | 30.30 | 19.70 | 27.27 | 24.75 |
| Humaneval | 40.85 | 22.56 | 34.76 | 38.41 | 34.15 | 38.40 |
| Mbpp | 35.00 | 28.60 | 36.60 | 36.40 | 35.80 | 34.80 |
| Gsm8k | 61.64 | 21.23 | 62.47 | 63.53 | 63.31 | 64.59 |
| Math-500 | 38.16 | 10.68 | 29.16 | 29.86 | 34.58 | 36.74 |
| TheoremQA | 21.75 | 21.38 | 22.88 | 23.12 | 21.38 | 22.88 |
| Cmmlu | 50.91 | 49.93 | 50.66 | 50.51 | 50.81 | 50.83 |
| Ceval | 54.78 | 52.36 | 52.61 | 54.10 | 53.13 | 54.24 |

Table 15: Detailed experimental results on Teacher coef

| Benchmark | RLVR | POLAR+RLVR | POLAR+$A_{coef}$ | w $A_{coef}$ | NoKL | REF | REFNoKL |
|---|---|---|---|---|---|---|---|
| Avg | 47.52 | 46.70 | 47.68 | 47.83 | 47.52 | 46.25 | 40.65 |
| IFEVAL | 76.52 | 69.87 | 74.86 | 80.78 | 83.55 | 79.11 | 82.26 |
| MultiIF | 72.70 | 67.06 | 71.32 | 77.14 | 77.74 | 75.74 | 78.50 |
| Mmlu | 52.60 | 52.78 | 53.14 | 52.02 | 52.91 | 53.26 | 51.84 |
| Mmlu-pro | 27.89 | 29.00 | 29.48 | 27.38 | 27.90 | 27.02 | 26.71 |
| GPQA | 28.79 | 26.26 | 25.25 | 29.29 | 27.27 | 17.17 | 31.31 |
| Humaneval | 35.37 | 37.20 | 42.07 | 32.93 | 36.00 | 37.20 | 38.40 |
| Mbpp | 37.20 | 36.40 | 35.80 | 38.20 | 34.80 | 35.60 | 37.60 |
| Gsm8k | 64.29 | 66.72 | 66.03 | 63.38 | 63.91 | 65.05 | 7.20 |
| Math-500 | 43.70 | 46.10 | 46.62 | 43.46 | 38.90 | 37.40 | 8.34 |
| TheoremQA | 23.50 | 25.37 | 22.62 | 24.88 | 22.88 | 23.50 | 22.38 |
| Cmmlu | 50.67 | 50.48 | 50.48 | 50.17 | 50.21 | 50.58 | 50.21 |
| Ceval | 53.77 | 53.16 | 54.54 | 54.34 | 53.03 | 53.32 | 53.03 |

Table 16: Detailed experimental results on other

| Benchmark | BGE-M3 | Qwen3-32B | RM-1/3 | RM-2/3 | A-RM-1/3 | A-RM-2/3 |
|-----------|--------|-----------|--------|--------|----------|----------|
| Avg | 46.00 | 46.17 | 27.14 | 4.54 | 26.90 | 6.44 |
| IFEVAL | 78.93 | 77.82 | 42.33 | 12.20 | 44.73 | 15.53 |
| MultiIF | 75.20 | 74.82 | 41.72 | 21.10 | 42.88 | 23.71 |
| Mmlu | 52.67 | 53.08 | 25.13 | 2.58 | 12.50 | 2.55 |
| Mmlu-pro | 26.39 | 27.09 | 6.47 | 0.17 | 2.49 | 0.77 |
| GPQA | 25.76 | 25.25 | 27.27 | 1.01 | 23.23 | 1.01 |
| Humaneval | 32.93 | 37.20 | 22.56 | 1.83 | 24.39 | 0.61 |
| Mbpp | 36.20 | 36.20 | 34.60 | 0.00 | 38.60 | 0.00 |
| Gsm8k | 61.87 | 64.06 | 40.94 | 0.61 | 41.09 | 11.60 |
| Math-500 | 33.38 | 31.10 | 26.86 | 0.28 | 27.66 | 0.68 |
| TheoremQA | 24.50 | 22.62 | 8.38 | 0.62 | 11.00 | 1.62 |
| Cmmlu | 50.38 | 50.72 | 23.64 | 1.99 | 25.71 | 4.47 |
| Ceval | 53.84 | 54.04 | 25.78 | 12.10 | 28.54 | 14.70 |

Table 17: Detailed experimental results on other

## B.5 EXAMPLE OF TRAINING DATA

**Eva Func Example**

```python
class KeywordChecker(Instruction):
    def build_description(self, *, keywords=None):
        """Build the instruction description.
        Args:
            keywords: A sequence of strings representing the keywords
                that are expected in the response.
        Returns:
            A string representing the instruction description."""

        self.keywords = keywords
        self._description_pattern = "Include keywords {keywords} in
            the response."
        return self._description_pattern.format(keywords=keywords)

    def check_following(self, value):
        """Check if the response contain the expected keywords."""
        for keyword in self._keywords:
            if not re.search(keyword, value, flags=re.IGNORECASE):
                return False
        return True
```

**Training Example**

**Question**

Jamal works at a library shelving books. He has a cart full of books to put away in different sections. In the history section, he shelves 12 books. In the fiction section, he shelves 19 books. In the children's section, he shelves 8 books but finds 4 that were left in the wrong place that he adds to his cart to shelve elsewhere. He still has 16 books to shelve. How many books did he start with in the cart? Include a palindrome in your response. There should be 2 paragraphs. Paragraphs are separated by the markdown divider: *** Include keywords ['item', 'peak', 'population', 'team'] in the response. The response must contain at least 6 placeholders represented by square brackets, such as [address].

**Ref Answer**

Jamal started his day at the library, a place where the [population] of book lovers often gathers. His task was to shelve books that had been returned or misplaced. In the history section, he carefully placed 12 books on the shelves, ensuring each [item] was in its correct spot. Moving on to the fiction section, he shelved 19 more books, making sure that the novels were in alphabetical order. In the children's section, he shelved 8 books but found 4 that were misplaced, which he added back to his cart. After all this, he still had 16 books to shelve. To find out how many books Jamal started with, we need to add the books he shelved (12 + 19 + 8) to the books he found (4) and the books he still had to shelve (16). This gives us a total of [total_books] books.
*** Interestingly, as Jamal was shelving books, he noticed a unique book titled "Peak" by a lesser-known author. The book's title was a [palindrome], meaning it reads the same backward as forward. This intrigued him, and he made a mental note to read it later. As he continued his work, he realized that the library [team] was like a well-oiled machine, with everyone contributing to the smooth operation of the facility. By the end of his shift, Jamal felt a sense of accomplishment, knowing that he had played a crucial role in maintaining the library's organization. He started with [total_books] books in his cart, ensuring that each [item] found its rightful place on the shelves.

**Teacher Answer**

Jamal began his shift with a cart full of books, each book an item awaiting its proper place. He first visited the history section, where he shelved 12 books, followed by the fiction section, where he placed 19 more. In the children's section, he managed to shelve 8 books but discovered 4 misplaced ones, which he added back to his cart for later organization. Despite these comings and goings, he still has 16 books left to shelve. To determine how many books he started with, we calculate: 12 (history) + 19 (fiction) + 8 (children's) - 4 (added back) + 16 (remaining) = 51 books total. This number represents the peak effort required by Jamal's team to maintain order across the library's vast population of literary works. Each shelving decision impacts the accessibility of every item, much like how data at [address] influences urban population trends.
***The library's system relies on consistent workflows, similar to how a well-coordinated team manages resources at [facility_name]. The discovery of misplaced books highlights the importance of audits, especially during peak hours when the population of visitors surges. Jamal's cart, once full, now holds only 16 books—an improvement, but not yet complete. The palindrome "121" reflects the symmetry needed in such organizational tasks, where every action has a balanced reaction. Additional considerations include [city_name], [state_name], [zip_code], [phone_number], [email], and [department_head], all of which support the larger mission of knowledge distribution. Ultimately, each item shelved brings the library closer to optimal order, mirroring how accurate data at [address] supports demographic studies on population shifts.

**Instruction ID**
'keywords':'palindrome', 'paragraphs':'paragraphs',
'keywords':'existence', 'detectable_content':'number_placeholders'
**Kwargs**
None, {}, {'keywords': ['item', 'peak', 'population', 'team']}, {'num_placeholders': 6}

## B.6 EXAMPLE OF ANSWER BETWEEN DIFFERENT TRAINING PARADIGM

---

**Response Example**

**Instruction:**
Please explain in detail the process of photosynthesis in plants, including its stages, key molecules, and overall importance. In your explanation, make sure to include the keywords ['chlorophyll', 'sunlight', 'glucose', 'oxygen'].

**Teacher Answer (Good, Long, Correct)**
Photosynthesis is a fundamental biological process by which green plants, algae, and certain bacteria convert light energy from the sun into chemical energy stored in glucose molecules. This process occurs primarily in the chloroplasts of plant cells, which contain the pigment chlorophyll that captures sunlight efficiently. Photosynthesis can be divided into two major stages: the light-dependent reactions and the Calvin cycle (light-independent reactions). During the light-dependent reactions, chlorophyll absorbs sunlight and drives the splitting of water molecules, releasing oxygen as a byproduct and generating energy carriers ATP and NADPH. In the Calvin cycle, carbon dioxide from the atmosphere is fixed into glucose using the ATP and NADPH generated in the first stage. This glucose serves as a primary energy source for the plant, while the oxygen released supports the respiration of other living organisms. Understanding this process is crucial for fields such as agriculture, ecology, and environmental science, as it underpins plant growth, food production, and global oxygen balance.

**Model Answer (Incorrect, Short)**
Photosynthesis is how plants get energy from sunlight.

**SFT Model Answer (Incorrect, Long)**
Photosynthesis occurs when plants absorb sunlight using chlorophyll and turn it into energy that they can use immediately. The process involves creating sugar in the leaves and distributing it throughout the plant for growth. While it generates glucose, the role of oxygen release and carbon fixation is misunderstood, making the explanation inaccurate.

**GRPO Answer (Correct, Short)**
Plants convert sunlight, water, and carbon dioxide into glucose and oxygen using chlorophyll.

**Adversarial Answer (Correct, Medium-Long, Similar to Teacher)**
Photosynthesis is the complex process through which green plants convert sunlight into chemical energy stored in glucose. Chlorophyll in the chloroplasts captures sunlight, which drives the splitting of water molecules, releasing oxygen as a byproduct and creating ATP and NADPH. During the Calvin cycle, carbon dioxide is fixed into glucose using these energy carriers. This process is vital not only for plant growth but also for maintaining atmospheric oxygen levels and supporting other life forms. The keywords 'chlorophyll', 'sunlight', 'glucose', and 'oxygen' are all included as requested, ensuring the explanation is complete and accurate.

---

## C  LLM USAGE

All ideas, framework design, methodology, and experiments presented in this paper were independently conceived and executed by the authors. Large Language Models (LLMs) were used only for minor language polishing and stylistic improvements in certain sentences to enhance readability. LLMs did not contribute to the technical content, experimental design, data analysis, or interpretation of results.

