# OpenReview forum: "UniAPL: A Unified Adversarial Preference Learning Framework for Instruct-Following"
_ICLR.cc/2026/Conference — Submitted to ICLR 2026_

### Official Review · Reviewer_WXiZ · 2025-10-16

**Soundness:** 3
**Presentation:** 3
**Contribution:** 3
**Rating:** 6
**Confidence:** 2

**Summary:**

The paper introduces a single-stage unified adversarial preference learning framework (UniAPL) for LLM post-training. Unlike traditional multi-stage approaches, UniAPL integrates supervised fine-tuning (SFT) and reinforcement learning (RL) preference data within the same training batches. It further incorporates a distributional regularizer to ensure that each update simultaneously optimizes imitation learning, preference alignment, and consistency with the expert data distribution. The proposed approach demonstrates strong and consistent performance across multiple benchmarks.

**Strengths:**

The paper presents a precise, gradient-level formulation that clearly shows how SFT, RL, and distributional regularization interact at each update. This unified view is both conceptually clarifying and practically convenient.

By reusing POLAR, the method sidesteps the instability and computational cost of training a separate discriminator while still providing a strong distributional learning signal.

Solid performance

**Weaknesses:**

The POLAR-based discriminator remains fixed, making the setup feel closer to a reference-regularized objective than true co-trained adversarial learning. Clarifying whether is ever updated, and analyzing how this affects robustness under domain shift.

Since the teacher model is not guaranteed to be correct, UniAPL might inherit its biases or mistakes, particularly when adversarial and reward signals diverge. A stress test or controlled experiment could better demonstrate resilience in such cases.

The evaluation focuses mainly on instruction-following tasks. Including results on safety, harmlessness, or human preference settings would strengthen the broader alignment claims.

**Questions:**

See Weaknesses

---

> ### Author Response · Authors · 2025-11-22
> **Response to Reviewer  WXiZ  (Part 1)**
>
> We appreciate your recognition of our unified perspective and the experimental work. We will now address your comments point by point.
>
> **Weaknesses 1**
> >The POLAR-based discriminator remains fixed, making the setup feel closer to a reference-regularized objective rather than true co-trained adversarial learning. It would be helpful to clarify whether the discriminator is ever updated, and to analyze how this design choice affects robustness under domain shift.
>
> To avoid additional computational overhead, we do not update the discriminator during training, and we explicitly state this in the paper. Following the principle of simplicity as effectiveness, we adopt a modified variant of POLAR. Nevertheless, your suggestion is meaningful. Therefore, we include an additional ablation study where BGE-M3 is used as the discriminator. Although its performance is not as strong as POLAR, it is still effective. Below is a brief summary of the instruction-following capability; detailed results are provided in Appendix B.2.
>
> **Weaknesses2**
> > Since the teacher model is not guaranteed to be correct, UniAPL might inherit its biases or mistakes, particularly when adversarial and reward signals diverge. A stress test or controlled experiment could better demonstrate resilience in such cases.
>
> Indeed, the teacher model cannot be entirely correct. Therefore, in the appendix, we perform an ablation study using ground-truth answers. In addition, we include experiments with Qwen32B, which still outperform GRPO. Below is a brief summary of the instruction-following capability; detailed results can be found in Appendix B.2.
>
> | Model/setting    | Avg_IF     |   IFEVAL   |  MultiIF |
> |----------------------|-----------|-----------|-----------|
> |  BGE-M3  as Discriminator|    77.07   |    78.93       |    75.20    |
> | POLAR as Discriminator |    77.24   |     78.74   |   75.74   |
> | Qwen3-235B as Teacher(same) |   77.24   |   78.74        |    75.74       |
> | Qwen3-32B as Teacher   |    76.32  |    77.82    |    74.82   |

---

> ### Author Response · Authors · 2025-11-22
> **Response to Reviewer  WXiZ  (Part 2)**
>
> **Weaknesses3**
>
> >The evaluation focuses mainly on instruction-following tasks. Including results on safety, harmlessness, or human preference settings would strengthen the broader alignment claims.
>
> To validate the generality of our method, we add experiments on mathematics and reward-model scoring tasks. Both experiments confirm the effectiveness of our adversarial coefficient, though each was trained for only one epoch. Notably, reward-model scoring can easily favor “Reward Hacking” behaviors, but the adversarial gradient effectively mitigates this issue. Below is a brief summary of the instruction-following capability; detailed results are available in Appendix B.2. Additionally, the mathematics experiments are also included in Appendix B.2.
> | Internlm2-20b-Reward    | Avg_IF     |   IFEVAL   |  MultiIF |
> |------------------|-----------|-----------|-----------|
> | RM-1/3 epoch |   42.03     |   42.33     |   41.72    |
> | RM-2/3 epoch |    16.65       |   12.20   |   21.10  |
> | ARM-1/3 epoch   |   43.81    |   44.73   |  42.88  |
> | ARM-1/3 epoch   |   19.62   |   15.53  |   23.71    |
>
> | Method    | Avg     |   AIME24   |  AIME25| AMC|
> |------------------|-----------|-----------|-----------|-----------|
> |Qwen2.5-7B-Instruct|   20.72     |   11.7     |   6.66    | 43.80 |
> | Luffy |   23.69       |   10.00   |   14.16  | 46.92 |
> | CHORD  |  24.25  |   16.25   |  15.00  | 41.51 |
> | A-CHORD   |   24.64   |   15.53  |   16.25   | 42.26|

---

### Official Review · Reviewer_gyGb · 2025-10-23

**Soundness:** 3
**Presentation:** 2
**Contribution:** 3
**Rating:** 4
**Confidence:** 3

**Summary:**

The paper proposes UniAPL, a unified adversarial preference learning framework for LLM post-training. Instead of the standard two-stage SFT -> RL pipeline (which the authors argue suffers from a distributional mismatch between expert data and the evolving policy), UniAPL couples demonstrated preferences (SFT) and comparative preferences (RL) in a single-stage objective. Concretely, a discriminator produces an adversarial signal that pulls the student’s outputs toward the teacher/expert distribution during both SFT and RL steps. The unified loss mixes adversarial SFT and adversarial GRPO on the same training run. Empirically, using AutoIF/IFevallike/IFBench with verifiable rewards and Qwen3-235B-Instruct-2507 as the teacher, UniAPL improves instruction-following while maintaining or improving general capabilities

**Strengths:**

1. Recasts alignment as a constrained optimization with an explicit expert-faithfulness constraint and identifies the distributional mismatch from sequential SFT -> RL as the core issue.
2. The paper spells out the unified loss and decomposes the update into imitation, preference, and global distributional regularization terms.
3. Strong empirical results on instruction-following with competitive general capability retention; includes staged and unified settings and ablations showing the adversarial term works best as a separate loss (not reward shaping).

**Weaknesses:**

1. The discriminator is adapted from POLAR and the teacher is Qwen3-235B; robustness to other teachers/discriminators is not evaluated. This matters because the “semantic manifold” you match could bias exploration if the teacher has idiosyncrasies.
2. The paper reports staged pipelines (SFT -> GRPO, SFT -> A-GRPO, A-SFT -> A-GRPO) but never varies when the switch happens (e.g., early vs. late switch, partial SFT). It's unclear whether the staged pipelines are switched optimally.
3. Results are presented as end-of-training tables and a few static analyses (e.g., response-length alignment), but there are no training curves for reward/pass-rate, KL, or the adversarial term over time.
4. Implementation details list learning rates for SFT, GRPO, and mixed training but do not state the number of steps/epochs, early-stopping rules, eval cadence, or total tokens, so it’s unclear how SFT and RL were decided to stop (fixed budget vs. dev-set early stop).

**Questions:**

1. How do results change with a different teacher (e.g., a smaller or domain-specialized instructor)? Could UniAPL still outperform its teacher on IFEval under such swaps?
2. Do you observe reduced exploration (e.g., narrower diversity in sampled candidates) when $\lambda_{adv}$ is higher? Any metrics on entropy or solution diversity during RL?
3. The unified run simplifies workflow, but what is the wall-clock/compute comparison vs. the best-tuned two-stage baseline at matched final quality? A cost-quality curve would be informative.
4. What stopping criteria did you use for SFT, RL, and unified training?

---

> ### Author Response · Authors · 2025-11-22
> **Response to Reviewer  gyGb (Part 1)**
>
> We sincerely appreciate your recognition of our paper and experiments. We will now respond to your comments point by point.
>
> **Weakness 1 & Question 1: Robustness to different teachers and discriminators**
> > 1.The discriminator is adapted from POLAR and the teacher is Qwen3-235B; robustness to other teachers/discriminators is not evaluated. This matters because the “semantic manifold” you match could bias exploration if the teacher has idiosyncrasies.
> > 1.How do results change with a different teacher (e.g., a smaller or domain-specialized instructor)? Could UniAPL still outperform its teacher on IFEval under such swaps?
>
> We provide ablation experiments using the dataset’s built-in standard answers in Appendix B.2. However, your comment is very meaningful, so we **added Qwen3-32B as the teacher** in an ablation study. The 4B model trained with AGRPO still **outperforms the 32B teacher**. Additionally, we include an ablation with **BGE-M3 as the discriminator**, which still achieves better performance. This validates the **effectiveness of the adversarial gradient**.
>
> Below we summarize these two ablations on the instruction-following task (detailed results are in the updated Appendix B.2):
>
> | Model/setting    | Avg_IF     |   IFEVAL   |  MultiIF |
> |----------------------|-----------|-----------|-----------|
> |  BGE-M3  as Discriminator|    77.07   |    78.93       |    75.20    |
> | POLAR as Discriminator |    77.24   |     78.74   |   75.74   |
> | Qwen3-235B as Teacher(same) |   77.24   |   78.74        |    75.74       |
> | Qwen3-32B as Teacher   |    76.32  |    77.82    |    74.82   |

---

> ### Author Response · Authors · 2025-11-22
>
> **Weakness 2 & Weakness 4 & Question 4: Experimental fairness in staged vs. unified training**
> > 2.The paper reports staged pipelines (SFT -> GRPO, SFT -> A-GRPO, A-SFT -> A-GRPO) but never varies when the switch happens (e.g., early vs. late switch, partial SFT). It's unclear whether the staged pipelines are switched optimally.
>
> > 4.Implementation details list learning rates for SFT, GRPO, and mixed training but do not state the number of steps/epochs, early-stopping rules, eval cadence, or total tokens, so it’s unclear how SFT and RL were decided to stop (fixed budget vs. dev-set early stop).
>
> > 4.Do you observe reduced exploration (e.g., narrower diversity in sampled candidates) when $\lambda_{adv}$  is higher? Any metrics on entropy or solution diversity during RL?
>
> In the staged experiments, SFT is run for 1 epoch followed by 1 epoch of RL. Under the unified view, using the same SFT and RL data, RL uses 8 rollouts, and the ratio of SFT to RL data per step is 1:4. Therefore, SFT effectively runs for 2 epochs while RL runs for 1 epoch. This could affect fairness.
>
> We clarify that we did not early-stop any experiments, which could affect the comparability between staged and unified training. We have **added detailed explanations in Appendix B.1** to ensure transparency regarding this experimental setup.
>
> **Weakness 3 & Question 3: Computational overhead of adversarial gradients**
> > 3.Results are presented as end-of-training tables and a few static analyses (e.g., response-length alignment), but there are no training curves for reward/pass-rate, KL, or the adversarial term over time.
>
> > 3.What is the additional computational cost of incorporating adversarial gradients?
>
> According to our theory, the overhead mainly arises from two sources, but it is **negligible in the context of reward-model scoring**, aligning with our Occam’s razor principle of avoiding real-time updates:
>
> 1. **Loss summation:** We introduce a scalar multiplied by the log-likelihood, which incurs almost **no extra cost during actor updates**. We include a comparison of actor update time for the 4B model in **Appendix B.3**, showing minimal change.
> 2. **Discriminator evaluation:** Typically deployed as a network service, discriminators in RLHF are smaller than reward models and can be called via multithreading **without notable overhead**. In RLVR, the discriminator cost depends on inference speed. A-SFT requires additional rollouts, which are computationally expensive. Therefore, we **recommend integrating adversarial gradients into A-GRPO or unified training** rather than directly using A-SFT. This rationale is detailed in Appendix B.3.
>
> Additionally, beyond static model evaluation, we provide **reward comparisons at the first 100 steps of 4B model training** in Appendix B.3. This shows that models with adversarial gradients achieve slightly higher rewards, connecting overhead analysis with performance.

---

> ### Comment · Reviewer_gyGb · 2025-11-23
>
> Thank you for your detailed response. Your answer partly addressed my concern, but I have one remaining question.
>
> Conceptually, your approach seems very similar to [1], where, instead of using the discriminator proposed in your paper, reverse KL divergence is directly used for distillation. How does your method differ from [1]? Is there a reason why you did not discuss or compare against [1]?
>
> I understand that adding an additional experiment at this late stage of the rebuttal phase may be too much to ask. Any other form of clarification or justification that clearly differentiates your method from [1] would also be sufficient. That said, if it is feasible, a quick empirical comparison would certainly help.
>
> If you can clearly demonstrate sufficient differentiation from [1], I will raise my score.
>
> ---
>
> Related Work
>
> [1] On-Policy Distillation of Language Models: Learning from Self-Generated Mistakes, ICLR 2024
>
> [2] On-Policy Distillation, https://thinkingmachines.ai/blog/on-policy-distillation/, Thinking Machines, 2025

---

> ### Author Response · Authors · 2025-11-27
>
> Thank you again for your insightful suggestions. We would like to clarify that GKD remains a token-level feedback distillation method, where distribution matching is performed through reverse KL between the student and teacher. Below we provide both a theoretical and implementation-level comparison to highlight our differences, followed by an initial rapid experiment, and we are actively running full experiments. **If necessary**, we will include the additional results in the supplementary ablations.
>
> 1. **Theoretical perspective.**
>    GKD relies on the teacher model’s *token-level* log-probabilities, which represents the core difference between the two approaches. From the SFT viewpoint, token-level learning is still subject to *knowledge brittleness* — the issue described in the second link you provided, where IFEVAL drops significantly after imitation. From the RL viewpoint, GKD behaves similarly to a *teacher-logprob-dependent PRM process*. In contrast, our adversarial mechanism operates at the *sentence-level*, where the discriminator produces a holistic preference score that applies uniformly to all tokens. This avoids overly local token imitation and focuses instead on global response similarity.
>
> 2. **Implementation perspective.**
>    In ms-swift, the implementation you referenced is `GKDTrainer`, and the JSD loss experiments rely on the teacher’s logits (**white-box access**). Our adversarial gradient is fundamentally different: the discriminator is *task-oriented and model-agnostic*, meaning that we do not require online teacher inference or log-probabilities. We only need the teacher’s output text *offline* (**black box**), which can even come from stronger closed-source models (GPT / Gemini).
>    When applied in RL, our primary objective remains maximizing the reward through self-improvement of the policy. When applied in SFT or offline objectives (DPO, KTO), the signal also remains sentence-level and global.
>
> To ensure fairness, we conducted a rapid experiment using the same offline teacher data from **Qwen3-32B** . The training followed the **ms-swift offline GKD pipeline**¹ (first inference, λ = 0). In parallel, we are running full Qwen3-32B online training (λ = 1, β = 1) and ASFT training. We observe that full online distillation is extremely slow — estimated to be **20× slower than SFT** (3 hours vs 3 days on 4×H100), and even slower than GRPO with rollout=8, because it requires real-time teacher inference (training with 235B would be even more expensive).
> Out of academic curiosity, we will still complete the full run and analyze the results. Preliminary findings already indicate that the offline GKD experiments have achieved performance surpassing that of SFT.
> | Model | Avg_general | Avg_IF | IFEVAL | MultiF |
> |-------|-------------|--------|--------|--------|
> | Qwen3-0.6B | 40.81 | 56.62 | 55.64 | 57.60 |
> | SFT (32B) | 39.61 | 58.35 | 58.96 | 57.74 |
> | ASFT (235B) | 39.83 | 60.72 | 61.74 | 59.70 |
> | GKD_offline (32B) | 38.09 | 59.82 | 61.74 | 57.89 |
>
> > ¹ https://swift.readthedocs.io/zh-cn/latest/Instruction/GKD.html

---

> ### Comment · Reviewer_gyGb · 2025-11-27
>
> Thanks for your detailed response. Your response fully addresses my concerns. I have increased my score. Good luck.

---

> > ### Author Response · Authors · 2025-11-27
> >
> > Thank you for your generous scoring. Your earlier questions regarding the teacher model and experimental details have been immensely helpful in refining this paper. Moreover, your observation about the distinction between our method and GKD has provided valuable insight into the fundamental differences between the approaches, which has greatly stimulated our deeper reflection. We will incorporate a more thorough analysis and further revisions into the paper accordingly.

---

### Official Review · Reviewer_XNu4 · 2025-10-26

**Soundness:** 2
**Presentation:** 2
**Contribution:** 2
**Rating:** 4
**Confidence:** 5

**Summary:**

This paper addresses the fundamental limitations of the standard sequential alignment pipeline (SFT-then-RL). The authors argue that this approach creates a critical distributional mismatch: SFT leads to "Imitative Brittleness" by overfitting to static expert data, while subsequent RL suffers from "Ungrounded Exploration," exploring without continuous access to the expert knowledge base.

**Strengths:**

The paper provides a conceptualization of the alignment challenge. The identification of "Imitative Brittleness" and "Ungrounded Exploration" as consequences of the distributional mismatch in sequential training is insightful. The proposed solution—a unified framework based on constrained optimization (Lemma 1)—is theoretically elegant.

The core mechanism of UniAPL, formalized in the unified gradient, ensures that every parameter update simultaneously incorporates signals for imitation, preference-seeking, and distributional grounding. This concurrent optimization simplifies the alignment workflow and allows SFT data to actively regularize RL exploration.

**Weaknesses:**

The core innovation of UniAPL is the adversarial regularization. However, the implementation deviates significantly from standard adversarial training (like GANs). The authors explicitly "avoid introducing a discriminator that would be progressively updated" (L359-360) and instead use a fixed, modified version of POLAR.9 This is a critical limitation. A fixed discriminator may be easily exploited by the evolving policy, or it may provide a static regularization signal that becomes ineffective as the policy distribution shifts. This undermines the claim of creating a "dynamic bridge" (L18) and raises questions about the robustness of the adversarial regularization.

The experimental validation is conducted exclusively on instruction-following tasks that utilize verifiable rewards (a binary reward based on a verification function; L364). This setting (RLVR) represents an idealized scenario with dense, perfect feedback. It is significantly different from general alignment (RLHF/RLAIF), where rewards are learned from noisy human preferences and ground truth is unavailable. The generalization of UniAPL to broader, subjective alignment challenges remains entirely unproven.

There is an inherent tension between the adversarial loss (enforcing mimicry of the teacher) and the RL objective (exploring to improve upon the base policy). Strong regularization might stifle exploration. The ablation studies (Figs. 5 and 6) confirm this sensitivity, showing significant performance degradation when the adversarial coefficient is set too high. This indicates potential instability and challenges in tuning the framework.

**Questions:**

In addition to the weakness section, I have the below questions:

Can you clearly distinguish UniAPL from existing Adversarial Imitation Learning frameworks like GAIL? What is the specific novelty beyond combining AIL with GRPO in a single stage?

How does UniAPL balance the constraint of the adversarial loss (mimicking the teacher) with the need for RL exploration? If the preference data suggests a solution that diverges significantly from the teacher's distribution, how does the unified objective handle this conflict?

---

> ### Author Response · Authors · 2025-11-22
> **Response to Reviewer XNu4 (Part 1)**
>
> Thank you for acknowledging the elegance of our theoretical framework. We will now address your comments point by point.
>
> **Weakness 1: Fixed discriminator vs. standard adversarial training**
> > The core innovation of UniAPL is the adversarial regularization. However, the implementation deviates significantly from standard adversarial training (like GANs). The authors explicitly "avoid introducing a discriminator that would be progressively updated" (L359-360) and instead use a fixed, modified version of POLAR.9 This is a critical limitation. A fixed discriminator may be easily exploited by the evolving policy, or it may provide a static regularization signal that becomes ineffective as the policy distribution shifts. This undermines the claim of creating a "dynamic bridge" (L18) and raises questions about the robustness of the adversarial regularization.
>
> Unlike GANs, avoiding a progressively updated discriminator is one of our design goals. In small-model alignment training, the parameters are limited, and updating the discriminator progressively would introduce significant computational overhead, contradicting the principle of Occam’s razor. We adopt POLAR because it has been trained across a wide variety of policy differences, which aligns better with our theoretical motivation. In our supplementary experiments, we include an ablation study with BGE-M3 in appendix B.2, showing that the approach still outperforms pure GRPO. Our main contribution is to demonstrate the **effectiveness of adversarial gradients**, which can dynamically regularize model responses during training. Regarding the "dynamic bridge," the nature is similar to reward models: different tasks can use different judgment signals to provide adversarial guidance.
>
> **Weakness 3: Tension between adversarial loss and RL exploration**
> > There is an inherent tension between the adversarial loss (enforcing mimicry of the teacher) and the RL objective (exploring to improve upon the base policy). Strong regularization might stifle exploration. The ablation studies (Figs. 5 and 6) confirm this sensitivity, showing significant performance degradation when the adversarial coefficient is set too high. This indicates potential instability and challenges in tuning the framework.
>
> It is true that exploration and adversarial regularization can interfere with each other. Our goal is to demonstrate that **adversarial gradients are effective** and can be added to any post-training module. The adversarial coefficient, like the KL coefficient, is a hyperparameter that requires tuning for different tasks. In our paper, we perform sensitivity ablations on the instruction fine-tuning task, and additionally include ablation experiments with the reward model and math tasks, all validating the effectiveness of the adversarial signal.
>
> **Question 1: Relation to GAIL**
> > Can you clearly distinguish UniAPL from existing Adversarial Imitation Learning frameworks like GAIL? What is the specific novelty beyond combining AIL with GRPO in a single stage?
>
> GAIL unifies adversarial training with traditional RL. In our work, we unify multiple post-training methods including GRPO, SFT, and DPO with adversarial training and apply it to large language models.
>
> **Question 2: Dynamic adjustment of adversarial coefficient**
> > How does UniAPL balance the constraint of the adversarial loss (mimicking the teacher) with the need for RL exploration? If the preference data suggests a solution that diverges significantly from the teacher's distribution, how does the unified objective handle this conflict?
>
> As discussed in Weakness 3, the adversarial coefficient currently requires manual tuning for different tasks, similar to the KL divergence coefficient. We have not yet implemented an automatic adjustment method, but this is an interesting direction for future research.

---

> ### Author Response · Authors · 2025-11-22
> **Response to Reviewer XNu4 (Part 2)**
>
> **Weakness 2: Generalization beyond instruction-following tasks**
> > The experimental validation is conducted exclusively on instruction-following tasks that utilize verifiable rewards (a binary reward based on a verification function; L364). This setting (RLVR) represents an idealized scenario with dense, perfect feedback. It is significantly different from general alignment (RLHF/RLAIF), where rewards are learned from noisy human preferences and ground truth is unavailable. The generalization of UniAPL to broader, subjective alignment challenges remains entirely unproven.
>
> We have supplemented experiments using **preference reward models**, where we use the **InternLM2-Reward-20B** model to score responses with the same instruction-following data. Although the reward score increases during training, the model can quickly "reward hack" after one epoch, leading to performance collapse. By adding **adversarial gradients**, this performance degradation is effectively mitigated. Below is a summary of instruction-following capability. `RM` denotes using the reward model only, and we report metrics at 1/3 and 2/3 of an epoch,The detailed metrics for other capabilities can be found in Appendix B.2.
>
>
> | Internlm2-20b-Reward    | Avg_IF     |   IFEVAL   |  MultiIF |
> |------------------|-----------|-----------|-----------|
> | RM-1/3 epoch |   42.03     |   42.33     |   41.72    |
> | RM-2/3 epoch |    16.65       |   12.20   |   21.10  |
> | A-RM-1/3 epoch   |   43.81   |   44.73   |  42.88  |
> | A-RM-1/3 epoch   |   19.62  |   15.53  |   23.71    |
>
> In addition, we have supplemented **mathematical task experiments**, which are detailed in Appendix B.2. These further validate the effectiveness of our approach across different tasks.

---

### Official Review · Reviewer_TS3p · 2025-10-27

**Soundness:** 2
**Presentation:** 3
**Contribution:** 3
**Rating:** 2
**Confidence:** 4

**Summary:**

This paper proposes a new training setup centered around adding an ‘adversarial gradient’ SFT and RL losses to guide the model to remain similar to the expert data distribution. They also propose a unified loss combining GRPO, SFT, and the adversarial gradient to unify post-training stages. Experiments on various evaluations show incorporating the adversarial gradient improves performance over just SFT or GRPO, and in unified settings that perform RL and SFT together.

**Strengths:**

- Unifying SFT and RL is an interesting direction, and the approach of adding a gradient that encourages the student to produce similar response to the teacher is interesting.
- The results seem to generally suggest that the additional gradient does indeed aid downstream performance without losing general abilities.

**Weaknesses:**

- I found the paper quite unclear to read, in a number of places:
  - How are the results in Figure 2 achieved? I couldn’t find any mention of it in the text. It would also be useful to see results across all the benchmarks used in the paper, beyond IFEval, as high IFEval could just be a result of overfitting to the benchmark (as shown by [3]).
  - The authors argue that SFT is a special case of preference learning, but how does the adversarial gradient play a part in the unified view? Doesn’t the SFT loss already encourage the model to match the expert distribution? Lines 243-246 say it is doing some form of ‘global distributional regularisation’, but it would be useful to explain this further. As it is, it feels like an additional term without much extra motivation.
  - How does the unified view link to the later experiments? As far as I can tell, the main experiment where SFT and DPO are unified instead uses CHORD, prior work on unifying SFT and RL, instead of the proposed loss.
  - The bolding in Table 2 is sometimes incorrect: the highest average score for 0.6B size models is SFT -> A-GRPO, not A-SFT->A-GRPO.
  - The proposed method is not always better than baselines: CHORD outperforms A-CHORD on average in table 3, SFT outperforms A-SFT in Table 2 for the 4B model, and A-SFT -> A-GRPO consistently underperforms SFT -> A-GRPO. Additionally, the difference in performance between methods is often under a point on average, and as such it would be useful to have some idea of the statistical significance of these results.
- Incorporating SFT losses into RL losses itself is not particularly novel, and was proposed by InstructGPT [1]. Additionally, it would be good to compare to other baselines that combine SFT and RL such as LUFFY [2].
- The experimental setup feels a little odd: the authors only train on instruction following data, but then evaluate on broader evaluations like math and MMLU. Does this not mean that the results are also reflecting how well each method maintains the original model performance, while fitting to the training data? More justification for why a more general RL / SFT dataset is not used would be useful.
- Very minor, but “adversarial” is consistently misspelled as “adversaril” in the paper.

Overall, I think this paper needs major work to be accepted. Currently, it is written in an unclear manner, and its results do not seem to show that it performs significantly better than baselines.

[1] Ouyang, Long et al. “Training language models to follow instructions with human feedback.” ArXiv abs/2203.02155 (2022).

[2] Yan, Jianhao et al. “Learning to Reason under Off-Policy Guidance.” ArXiv abs/2504.14945 (2025).

[3] Pyatkin, Valentina et al. “Generalizing Verifiable Instruction Following.” ArXiv abs/2507.02833 (2025).

**Questions:**

See weaknesses.

---

> ### Author Response · Authors · 2025-11-22
> **Response to Reviewer TS3p (Part 1)**
>
> Thank you for the detailed comment! In response, we have conducted additional experiments and revised the paper
> ## Weakness 1
>
> ### Sub-topic 1
> > How are the results in Figure 2 achieved? I couldn’t find any mention of it in the text. It would also be useful to see results across all the benchmarks used in the paper, beyond IFEval, as high IFEval could just be a result of overfitting to the benchmark (as shown by [3]).
> 1. In addition to IFEval, we performed a round of dialogue validation on Multi-IF, and this has been described in the updated setting.
> 2. In the main experimental tables, we report the average over both benchmarks.
> 3. We replaced the main IFEval (Figure 2) with overall Instruction-Flowing results.
> 4. Detailed evaluation of Qwen3-series base models is included in the appendix B.4, and corresponding statements in the main figure and text have been corrected.
> 5. In the appendix, the reported metrics are highlighted in **bold** for clarity.
>
> We also summarize the detailed results across instruction following benchmarks for Qwen3-series and UniAPL models in the table below:
>
> | Model            | Avg_IF     |   IFEVAL   |  MultiIF |
> |-----------------|------------|------------|------------|
> | UniAPL-0.6B       |   79.86   |    82.99   |   76.73  |
> | UniAPL-4B         |   90.65   |    90.02    |  91.28     |
> | Qwen3-0.6B       |    56.62   |   55.64    |  57.60     |
> | Qwen3-1.7B       |    67.59  |    66.73   |   68.44    |
> | Qwen3-4B        |     80.10  |   78.19    |  82.00     |
> | Qwen3-4B-2507   |     82.84  |   80.59     | 85.09       |
> | Qwen3-8B        |     84.48  |   82.26    |   86.69     |
> | Qwen3-32B       |     85.04  |   84.29    |   85.79    |
> | Qwen3-235B-2507 |     89.18  |   87.99    |    90.37    |
> *Table: Detailed results across instruction following benchmarks for Qwen3-series and UniAPL-series models.*

---

> ### Author Response · Authors · 2025-11-22
> **Response to Reviewer TS3p (Part 2)**
>
> **Sub-topic 2: Role of adversarial gradient in the unified view**
> > The authors argue that SFT is a special case of preference learning, but how does the adversarial gradient play a part in the unified view? Doesn’t the SFT loss already encourage the model to match the expert distribution? Lines 243-246 say it is doing some form of ‘global distributional regularisation’, but it would be useful to explain this further. As it is, it feels like an additional term without much extra motivation.
>
> In the unified view, the student is treated as a policy and the teacher as a policy. The adversarial gradient allows the student model to receive gradient signals through scores, thereby bringing the student distribution closer to the teacher distribution. This provides a sentence-level unified signal. While SFT indeed encourages imitation, it essentially learns the teacher distribution. If the distributions differ significantly, SFT can fail. Moreover, SFT operates at the token-level and does not account for sentence-level similarity, capturing only superficial aspects, which is also validated by our analysis of response lengths.
>
> **Sub-topic 3: Link between the unified view and later experiments**
> > How does the unified view link to the later experiments? As far as I can tell, the main experiment where SFT and DPO are unified instead uses CHORD, prior work on unifying SFT and RL, instead of the proposed loss.
>
> The unified view aims to validate the effectiveness of sentence-level adversarial training. It can be applied to any post-training framework, whether staged or unified. In our experiments, we added our adversarial loss on top of CHORD to form A-CHORD, which distinguishes it from the original CHORD loss.
>
> **Sub-topic 4: Table 2 bolding errors**
> > The bolding in Table 2 is sometimes incorrect: the highest average score for 0.6B size models is SFT -> A-GRPO, not A-SFT->A-GRPO.
>
> Thank you for pointing this out! We have corrected the bolding to reflect the optimal results.
>
> **Sub-topic 5: Performance vs. baselines and statistical significance**
> > The proposed method is not always better than baselines: CHORD outperforms A-CHORD on average in table 3, SFT outperforms A-SFT in Table 2 for the 4B model, and A-SFT -> A-GRPO consistently underperforms SFT -> A-GRPO. Additionally, the difference in performance between methods is often under a point on average, and as such it would be useful to have some idea of the statistical significance of these results.
>
> Our goal is to show that adding a global adversarial term in specific tasks (instruction following, with supplementary math experiments in the appendix) can improve task performance while maintaining other capabilities, mitigating SFT knowledge fragility, rather than overfitting to a single benchmark. Accordingly, we also validated general abilities, which naturally fluctuate during training. Therefore, our method cannot guarantee that general ability scores are always optimal.

---

> > ### Comment · Reviewer_TS3p · 2025-11-23
> >
> > Thank you for the detailed response and large number of updates! I guess I still find the framing of this paper a little unusual, in which you propose this notion of a single unified loss in 3.4, but then primarily experiment with specific variants (A-SFT, A-GRPO), and staged training (for the 'main' UniAPL results), rather than some unified approach, which is also what you imply in your response ("[our approach] can be applied to any post-training framework, whether staged or unified"). I think some of the justifications in Appendix A.2 help with the framing, but they are not currently part of the main text.
> >
> > W1 Sub-topic 5: Do you have evidence of "mitigating SFT knowledge fragility" here? I get the arguments around response length distribution and logprobs, but it appears this has a much smaller effect on actual downstream performance (average diffs in performance between SFT/A-SFT, GRPO/A-GRPO, etc. don't seem that large in Table 2).
> >
> > I also think some of the claims from the new results are phrased a little strongly: claiming "the degradation in instruction-following ability is significantly mitigated" in B.2 is a bit much when you go from a 26-point drop to a 24-point drop.
> >
> > Weakness 3: Sorry, but what do you mean "[math evaluation] does not require additional training in Qwen3"? That you can evaluate without further training, that Qwen3 already does well at math without further training, or...?
> >
> > Another nit: rereading this paper, I notice there are number of times the spacing is missed (e.g., line 323, 345, 346, 888. 889).
> >
> > While I still think this paper needs some more polish, the response has addressed some of my concerns, so I'm raising my score.

---

> > > ### Author Response · Authors · 2025-11-27
> > >
> > > Thank you very much for your thorough follow-up! We have performed another round of lightweight revisions across the entire paper.
> > >
> > > **(1) On the framing of the unified loss and the staged training design**
> > > >Thank you for the detailed response and large number of updates! I guess I still find the framing of this paper a little unusual, in which you propose this notion of a single unified loss in 3.4, but then primarily experiment with specific variants (A-SFT, A-GRPO), and staged training (for the 'main' UniAPL results), rather than some unified approach, which is also what you imply in your response ("[our approach] can be applied to any post-training framework, whether staged or unified"). I think some of the justifications in Appendix A.2 help with the framing, but they are not currently part of the main text.
> > >
> > > To improve clarity, we have now added explicit references in the main text to every citation of Appendix A. For example, Section 3.4 now clearly directs readers to Appendix A.2.
> > >
> > > **(2) On whether we provide evidence of “mitigating SFT knowledge fragility”**
> > > >W1 Sub-topic 5: Do you have evidence of "mitigating SFT knowledge fragility" here? I get the arguments around response length distribution and logprobs, but it appears this has a much smaller effect on actual downstream performance (average diffs in performance between SFT/A-SFT, GRPO/A-GRPO, etc. don't seem that large in Table 2).
> > >
> > > The mitigation effect is primarily reflected in the instruction-following accuracy between A-SFT and SFT. Although the absolute improvement appears modest—A-SFT typically outperforms SFT by about 1%～2%—the relative gain is substantial, given the extremely limited room for improvement on the main SFT task。
> > > For example:
> > > - **0.6B:** (60.72 − 59.97) / (59.97 − 56.62) = **22% relative improvement**
> > > - **4B:** (82.47 − 80.60) / (80.60 − 80.10) = **374% relative improvement**
> > >
> > > Additionally, we conducted preliminary experiments (not included in the paper) scaling the magnitude of the ASFT adversarial loss. Using an adversarial coefficient on the same order of magnitude as the SFT loss proved counterproductive, as it caused the adversarial term to dominate the learning process. In contrast, using a coefficient 1–2 orders of magnitude smaller yielded consistently positive results.
> > >
> > > Although the SFT and A-SFT gap can appear mild, the improvement with **GRPO  and A-GRPO** is much more pronounced:
> > > - **0.6B:** 74.09 → 79.58
> > > - **4B:** 86.90 → 88.17
> > > As shown in Part 1 of  rebuttal, it is evident that once scores surpass the high-performance region (>80), achieving substantial further gains becomes exceedingly difficult.
> > >
> > >
> > > **(3) About the wording in Appendix B.2**
> > > >I also think some of the claims from the new results are phrased a little strongly: claiming "the degradation in instruction-following ability is significantly mitigated" in B.2 is a bit much when you go from a 26-point drop to a 24-point drop.
> > >
> > > We agree with your concern. The statement “the degradation in instruction-following ability is significantly mitigated” was softened in the revised version (remove "significantly") to make the description more moderate and objective.
> > >
> > > **(4) Clarification regarding “[math evaluation] does not require additional training in Qwen3”**
> > > >Weakness 3: Sorry, but what do you mean "[math evaluation] does not require additional training in Qwen3"? That you can evaluate without further training, that Qwen3 already does well at math without further training, or...?
> > >
> > > Sorry for the confusion — what we meant was specifically related to the **`<think>` format**. Mathematical reasoning tasks in evaluation typically include CoT-style thinking traces, and models normally require additional formatting supervision or system-prompt enforcement.
> > > However, Qwen3 provides built-in support for such formatting via the **`enable_thinking` flag**, so no additional format training is required (although this mechanism may sometimes produce empty `<think>` blocks). This behavior was also examined in the “think-label” ablation experiment in the initial version of Appendix B.2.
> > >
> > > **(5) Formatting issues**
> > > >Another nit: rereading this paper, I notice there are number of times the spacing is missed (e.g., line 323, 345, 346, 888. 889).
> > >
> > > Thank you for the very careful reading. We reviewed the entire manuscript again and corrected all spacing issues.

---

> > > > ### Comment · Reviewer_TS3p · 2025-11-27
> > > >
> > > > Thanks for the detailed updated response! The additional changes should help with clarity when reading the paper. Some additional notes below:
> > > >
> > > > 2. You say "The mitigation effect is primarily reflected in the instruction-following accuracy between A-SFT and SFT.", but in your previous response you said: "Our goal is to show that adding a global adversarial term in specific tasks (instruction following, with supplementary math experiments in the appendix) can improve task performance while maintaining other capabilities, mitigating SFT knowledge fragility, rather than overfitting to a single benchmark." I thought from this sentence that the mitigating was primarily referring to non-instruction following capabilities, since those are the ones you are explicitly training (and so fitting the most tightly to in training). In this sense, then the lack of movement of the average performance suggests that there isn't much difference in the amount of forgetting in SFT vs A-SFT? Or am I misunderstanding something here?
> > > >
> > > > 4. Thanks for this clarification! But then in the abstract you claim "models trained with UniAPL (not only) achieve stronger performance" - and your method seems pretty general. So why the focus on instruction-following, since your method should also work to improve math scores, as you demonstrate in the appendix? Sure, you don't need to teach the model new formats, but if your method is good, it should work for general math reasoning, code problems, etc, right?
> > > >
> > > > Finally, I also have a concern with the experiment in table 9 - only training for one epoch means that the scores are quite low relative to e.g. the numbers in LUFFY (LUFFY reports 50.9 in AMC, while Table 9 shows 46.9). The comparison is valid in the table, but I worry that if you trained for longer, the difference between methods wouldn't necessarily hold. That is, it may be the a-chord setup helps accelerate training a little, but doesn't necessarily improve performance over CHORD/LUFFY in higher-compute scenarios.

---

> > > > > ### Author Response · Authors · 2025-11-29
> > > > >
> > > > > Thank you for your feedback. We would like to further clarify these two points once again:
> > > > >
> > > > > >You say "The mitigation effect is primarily reflected in the instruction-following accuracy between A-SFT and SFT.", but in your previous response you said: "Our goal is to show that adding a global adversarial term in specific tasks (instruction following, with supplementary math experiments in the appendix) can improve task performance while maintaining other capabilities, mitigating SFT knowledge fragility, rather than overfitting to a single benchmark." I thought from this sentence that the mitigating was primarily referring to non-instruction following capabilities, since those are the ones you are explicitly training (and so fitting the most tightly to in training). In this sense, then the lack of movement of the average performance suggests that there isn't much difference in the amount of forgetting in SFT vs A-SFT? Or am I misunderstanding something here?
> > > > >
> > > > > SFT exhibits knowledge fragility in two main aspects. First, it often lacks sufficient generalization and merely learns surface-level patterns; the performance gain achieved by adding our adversarial gradient term suggests an improvement in generalization. Second, as you rightly pointed out, catastrophic forgetting is a concern. To address this, we evaluated general-domain performance to show that our method does not overfit to a single task and does not suffer from more severe catastrophic forgetting than standard SFT. As previously mentioned, our primary objective is to enhance task-specific performance while preserving general capabilities—hence, the extent of catastrophic forgetting between SFT and A-SFT is not expected to differ significantly.
> > > > >
> > > > > >Thanks for this clarification! But then in the abstract you claim "models trained with UniAPL (not only) achieve stronger performance" - and your method seems pretty general. So why the focus on instruction-following, since your method should also work to improve math scores, as you demonstrate in the appendix? Sure, you don't need to teach the model new formats, but if your method is good, it should work for general math reasoning, code problems, etc, right?
> > > > >
> > > > > >Finally, I also have a concern with the experiment in table 9 - only training for one epoch means that the scores are quite low relative to e.g. the numbers in LUFFY (LUFFY reports 50.9 in AMC, while Table 9 shows 46.9). The comparison is valid in the table, but I worry that if you trained for longer, the difference between methods wouldn't necessarily hold. That is, it may be the a-chord setup helps accelerate training a little, but doesn't necessarily improve performance over CHORD/LUFFY in higher-compute scenarios.
> > > > >
> > > > > As previously explained, instruction-following tasks serve not only as a benchmark for rule-based rewards but also provide a clearer and more discernible signal for evaluating methodological differences. Moreover, we disabled chain-of-thought (CoT) reasoning in our setup, meaning our responses do not involve lengthy CoT processes—this aligns with the typical length of instruction-following outputs, which are usually under 2K tokens. Importantly, our approach is designed to be general-purpose; accordingly, we have included additional evaluations using reward model scoring and mathematical reasoning tasks. We plan to continue training on the math tasks and incorporate those results into the paper. Finally, we are actively working on consolidating our method into a unified framework—currently it relies on three separate frameworks (Swift, VeRL, and Trinity)—and will open-source the code upon completion.

---

> ### Author Response · Authors · 2025-11-22
> **Response to Reviewer TS3p (Part 3)**
>
> **Weakness 2: Novelty of incorporating SFT into RL and baseline comparisons**
> > Incorporating SFT losses into RL losses itself is not particularly novel, and was proposed by InstructGPT [1]. Additionally, it would be good to compare to other baselines that combine SFT and RL such as LUFFY [2].
>
> Our approach differs from prior work in that we introduce a **global adversarial gradient term**, which provides an additional beneficial effect. Compared to LUFFY, we do not directly use teacher responses during the rollout process; all exploration is performed by the model itself, and we do not require any specialized shaping for off-policy expert responses. Nonetheless, your suggestion is reasonable, and we have supplemented our experiments with LUFFY in the mathematical tasks. Our method outperforms LUFFY in these experiments, and we also added preference reward model training results in Appendix B.2. Due to time constraints and the older LUFFY framework, we have not yet evaluated LUFFY on the instruction-following tasks.
>
> **Weakness 3: Experimental setup and dataset choice**
> > The experimental setup feels a little odd: the authors only train on instruction following data, but then evaluate on broader evaluations like math and MMLU. Does this not mean that the results are also reflecting how well each method maintains the original model performance, while fitting to the training data? More justification for why a more general RL / SFT dataset is not used would be useful.
>
> As noted in Weakness 1, Sub-topic 5, our primary goal is to evaluate the **instruction-following task**, where rule-based rewards are easier to define and the style differences between strategies are more visible. While math evaluation mainly serves as a final result check and `<think>` format verification, it does not require additional training in Qwen3. Our unified framework is designed to be computationally efficient, with minimal differences from standard online training. In contrast, ASFT introduces a rollout process, which significantly increases computational cost. Therefore, we did not adopt a broader SFT dataset. This point has been further emphasized in Appendix B.2 "Ablation on Different Task."
>
> **Weakness 4: Adversarial errors and spelling**
> > Very minor, but “adversarial” is consistently misspelled as “adversaril” in the paper.
>
> We have corrected all adversarial-related errors in the paper and carefully checked the text for spelling mistakes.

---

### Author Response · Authors · 2025-11-22
**Overall Response**

We sincerely thank the reviewers for recognizing the merit of our ideas, including the following aspects:
1. Appreciation for our adversarial framework for alignment (Reviewers Ts3p, XNu4, gyGb, and WXiZ).
2. Recognition of the clarity and interesting theoretical direction of introducing a standalone gradient term (Reviewers Ts3p, XNu4, and WXiZ).
3. Acknowledgment that the additional gradient can benefit downstream task performance (Reviewers Ts3p, gyGb, and WXiZ).

We would like to reiterate that our main contributions are:
1. Providing a unified perspective that frames SFT and RL as a single, coherent preference learning framework, enabling the use of the most appropriate algorithms to fully leverage heterogeneous data.
2. Introducing a global adversarial constraint gradient that brings the student and teacher models closer, demonstrating the broad effectiveness of the adversarial approach. This gradient can be integrated into any post-training framework, whether staged or unified.
3. We conducted detailed validation on instruction-following tasks using both the standard reference answers and the teacher-generated answers, evaluating Qwen-0.6B and Qwen-4B models. In the staged setup, the 0.6B model achieves a 5.77% improvement when combined with GRPO, while the 4B model shows a 3.75% gain. In addition, we analyzed the post-training responses of both the student and the teacher in terms of response length and output probabilities. The results indicate that incorporating adversarial training brings the student model’s preference responses closer to those of the teacher.

Finally, we conducted ablation studies on both the discriminator and the teacher-generated responses. In addition, we performed ablations on reward-model scoring and mathematical tasks to further validate the effectiveness and robustness of our approach.

---

### Author Response · Authors · 2025-11-29
**Final Remarks**

We sincerely thank all reviewers for their insightful comments, which have significantly improved the quality and completeness of this work. We are especially grateful to Reviewer TS3p and gyGb for their constructive dialogue across multiple rounds and for raising their scores (TS3p: 2 → 4, gyGb: 4 → 6).

Best regards,
The Authors

---

### Meta-Review · Area_Chair_8AD8 · 2025-12-28

**Summary:**

The authors propose a training framework that unifies SFT and RL through an adversarial loss with a fixed adversary. While this method improves performance, reviewers were generally concerned about:
(1) The positioning of the paper: it proposes a single unified training loss but its main experiments are still staged (e.g., SFT -> RL).
(2) Experimental design, especially around evaluations which focused on instruction following
(3) The choice of fixed adversary (vs. a dynamic adversary)

Review scores were initially low (2446) though they were revised upwards in the original rebuttal phase (2446 -> 4466). Overall, the revised scores are still borderline, and the core framing issues still remain. As the reviewers noted, this paper is interesting with the potential for impact, and I recommend that the authors revise and resubmit their work.

**Reviewer Concerns:**

See main comment.

**Reviewer Scores:**

In the original rebuttal phase, the scores moved from 2446 -> 4466. These would probaby have been similar if the full discussion had occurred.

---

### Decision · Program_Chairs · 2026-01-26

Reject